# Automatic deep learning-driven label-free image-guided patch clamp system

Krisztian Koos [1,7], Gáspár Oláh[2,7], Tamas Balassa[1], Norbert Mihut [2], Márton Rózsa[2], Attila Ozsvár[2], Ervin Tasnadi[1], Pál Barzó[3], Nóra Faragó[2,4,5], László Puskás[4,5], Gábor Molnár [2], József Molnár [1], Gábor Tamás [2] & Peter Horvath [1,6✉]

Patch clamp recording of neurons is a labor-intensive and time-consuming procedure. Here, we demonstrate a tool that fully automatically performs electrophysiological recordings in label-free tissue slices. The automation covers the detection of cells in label-free images, calibration of the micropipette movement, approach to the cell with the pipette, formation of the whole-cell configuration, and recording. The cell detection is based on deep learning. The model is trained on a new image database of neurons in unlabeled brain tissue slices. The pipette tip detection and approaching phase use image analysis techniques for precise movements. High-quality measurements are performed on hundreds of human and rodent neurons. We also demonstrate that further molecular and anatomical analysis can be performed on the recorded cells. The software has a diary module that automatically logs patch clamp events. Our tool can multiply the number of daily measurements to help brain research.

[1] Synthetic and Systems Biology Unit, Biological Research Centre, Eötvös Loránd Research Network, Szeged, Hungary. [2] MTA-SZTE Research Group for Cortical Microcircuits of the Hungarian Academy of Sciences, Department of Physiology, Anatomy and Neuroscience, University of Szeged, Szeged, Hungary. [3] Department of Neurosurgery, University of Szeged, Szeged, Hungary. [4] Laboratory of Functional Genomics, Institute of Genetics, Biological Research Centre, Szeged, Hungary. [5] Avidin Ltd, Szeged, Hungary. [6] Institute for Molecular Medicine Finland, University of Helsinki, Helsinki, Finland. [7] These authors contributed equally: Krisztian Koos, Gáspár Oláh. ✉email: horvath.peter@brc.hu

Research of the past decade uncovered the unprecedented cellular heterogeneity of the mammalian brain. It is well accepted now, that the complexity of the rodent and human cortex can be best resolved by classifying individual neurons into subsets by their cellular phenotypes[1–3]. By characterizing molecular, morphological, connectional, physiological, and functional properties several neuronal subtypes have been defined[4,5]. Revealing cell-type heterogeneity is still incomplete and challenging since classification based on quantitative features requires large amounts of individual cell samples, often thousands or more, encompassing a highly heterogeneous cell population. Recording morphological, electrophysiological, and transcriptional properties of neurons requires different techniques combined on the same sample such as patch clamp electrophysiology, posthoc morphological reconstruction, or single-cell transcriptomics. The fundamental technique to achieve such trimodal characterization of neurons is the patch clamp recording, which is highly laborious and expertise intense. Therefore, there is a high demand to efficiently automate this labor intense and challenging process.

Recently, the patch clamp technique has been automated and improved to a more advanced level[6,7]. Blind patch clamping was first done in vitro and only later performed in vivo[8–10]. In this case, the pipette is gradually moved forward and the brain cells are detected automatically by a resistance increase at the pipette tip. Automated systems soon incorporated image-guidance by using multiphoton microscopy on genetically modified rodents[11–13]. Further improvements include the integration of tools for monitoring animal behavior[14], the design of an obstacle avoidance algorithm before reaching the target cell[15] or the development of a pipette cleaning method which allows the immediate reuse of the pipettes up to ten times[16,17]. Automated multi-pipette systems were developed to study the synaptic connections[18,19]. It is also shown that cell morphology can be examined using automated systems[20]. One crucial step for image-guided automation is pipette tip localization. Different label-free pipette detection algorithms were compared previously[21]. Some automated patch clamp systems already contain pipette detection algorithms, e.g., intensity clustering[11] or thresholding-based[22] for fluorescence imaging, or Hough transform-based[23] for DIC optics. The other crucial step is the automatic detection of the cells which has only been performed in two-photon images so far. It is currently not possible to efficiently fluorescently stain human brain tissues. Alternatively, detection of cells in label-free images would open up new application possibilities in vitro[23], e.g., experiments on surgically removed human tissues. Most recently, deep learning[24] has been emerging to a level that in the case of well-defined tasks, outperforms humans, and often reaches human performance on ill-defined problems like detecting astrocyte cells[25].

In this paper, we describe a system we developed in order to overcome time-consuming and expertise-intense neuron characterization and collection. This fully automated differential interference contrast microscopy (DIC, or label-free in general) image-guided patch clamping system (DIGAP) combines 3D infrared video microscopy, cell detection using deep convolutional neural networks and a glass microelectrode guiding system to approach, attach, break-in, and record biophysical properties of the target cell.

The steps of the visual patch clamp recording process are illustrated in Fig. 1. Before the first use of the system, the pipette has to be calibrated, so that it can be moved relative to the field of view of the camera (1). Thereafter, a position update is made after every pipette replacement (2) using the built-in pipette detection algorithms (3) to overcome the problem caused by pipette length differences. At this point, the system is ready to perform patch clamp recordings. We have acquired and annotated a single cell

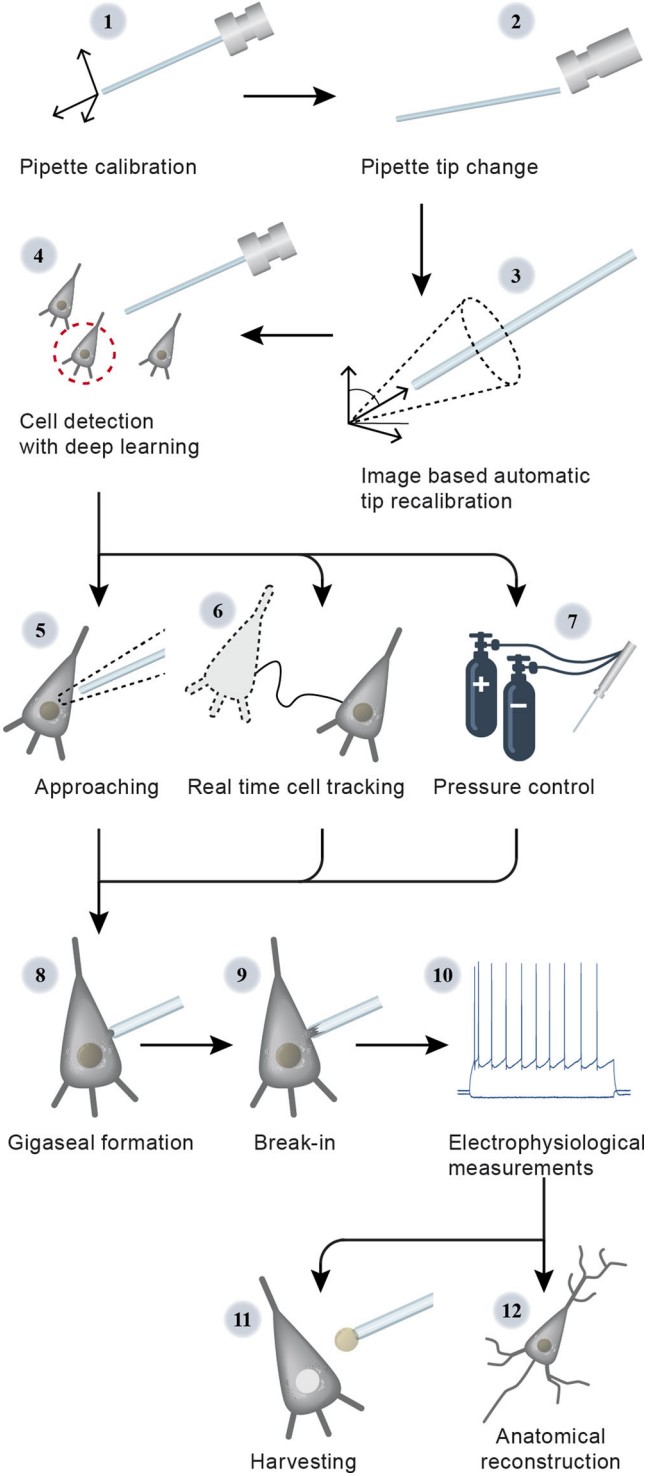

**Fig. 1 Steps of DIGAP procedures.** 1: Pipette calibration by the user, 2: pipette replacement after recording, 3: image-based automatic pipette tip detection, 4: automatic cell detection, 5: pipette navigation to the target cell, 6: 3D cell tracking, 7: pressure regulation, 8: gigaseal formation, 9: break-in, 10: electrophysiological recording, 11: nucleus and cytoplasm harvesting, 12: anatomical reconstruction of the recorded cell.

image database on label-free neocortical brain tissues, to our knowledge the largest 3D set of this kind. A deep convolutional neural network has been trained for cell detection. The system can automatically select a detected cell for recording (4). When a cell is selected, multiple subsystems are started simultaneously

that perform the patch clamping: (i) A subsystem controls the movement of the micropipette next to the cell. If any obstacle is found in the way, an avoidance algorithm tries to bypass it (5). (ii) A cell tracking system follows the possible shift of the cell in 3D (6). (iii) During the whole process, a pressure regulator system assures that the requested pressure on the pipette tip is available (7).

Once the pipette touches the cell (cell-attached configuration) the system performs gigaseal formation (8), then breaks in the cell membrane (9) and automatically starts the electrophysiological measurements (10). When the recording is completed, the operator can decide either to start over the process on a new target cell or continue with one or both of the following manual steps. The nucleus or the cytoplasm of the patched cell can be harvested (11), or the recorded cells can be anatomically reconstructed in the tissue (12).

At the end of the measurements, the implemented pipette cleaning method can be performed or the next patch clamp recording can be started after pipette replacement and from the pipette tip position update step (3). An event logging system collects information during the patch clamp process, including the target locations and the outcome success, and report files can be generated at the end. The report files are compatible with the Allen Cell Types Database[26].

Our system was tested on rodent and human samples in vitro. The quality of the electrophysiological measurements strongly correlates to that made by a trained experimenter. We have used the system for harvesting cytoplasm and nucleus from the recorded cells and performed anatomical reconstruction on the samples. Our system can operate on unstained tissues using deep learning, that reaches the cell detection accuracy of human experts, and that enables the multiplication of the number of recordings while preserving high-quality measurements.

## Results

Here, we introduce an automated seek-and-patch system that performs electrophysiological recordings and sample harvesting for molecular biological analysis from single cells on unlabeled neocortical brain slices. Using deep learning, trained on a previously built database of single neurons acquired in 3D, our system can detect most of the healthy neuronal somata in a Z-stack recorded by DIC microscopy from a living neocortical slice. The pipette approaches the target cell, touches it, acquires electrophysiological data, and the cell's nucleus can be isolated for further molecular analysis. Components of the system are a typical electrophysiological setup: IR video microscopy imaging system, motorized microelectrode manipulators, XY shifting table, electrical amplifier, and a custom-designed pressure controller. All these elements were controlled by a custom-developed software (available at https://bitbucket.org/biomag/autopatcher/). The system was successfully applied to perform patch clamp recordings on a large set of rodent and human cells (100 and 74, respectively). The automatically collected cells well represent the wide-range phenotypic heterogeneity of the brain cortex. Subsequent transcriptome profiling and whole-cell anatomical reconstruction confirmed the usefulness and applicability of the proposed system.

**Hardware development and control**. The hardware setup of the proposed system is shown in Fig. 2. The software system we developed controls each hardware using their drivers on application programming interface (API) level, which makes the system modular and different types of hardware components (e.g., manipulators, biological amplifier, and XZ shifting table) can be attached. The classes which control hardware elements are inherited from abstract classes. Thus, if the software is to be used with a different hardware element then only a few methods should be implemented in a child class that sends commands to that specific device (e.g., to get or set the pipette position or initiate a protocol in the amplifier's software).

The electrophysiological signal from the current monitor output of the amplifier is transferred to the DIGAP software via the analog input channel of the USB digitizer board (National Instruments, USB-6009), which enables real-time resistance measurement.

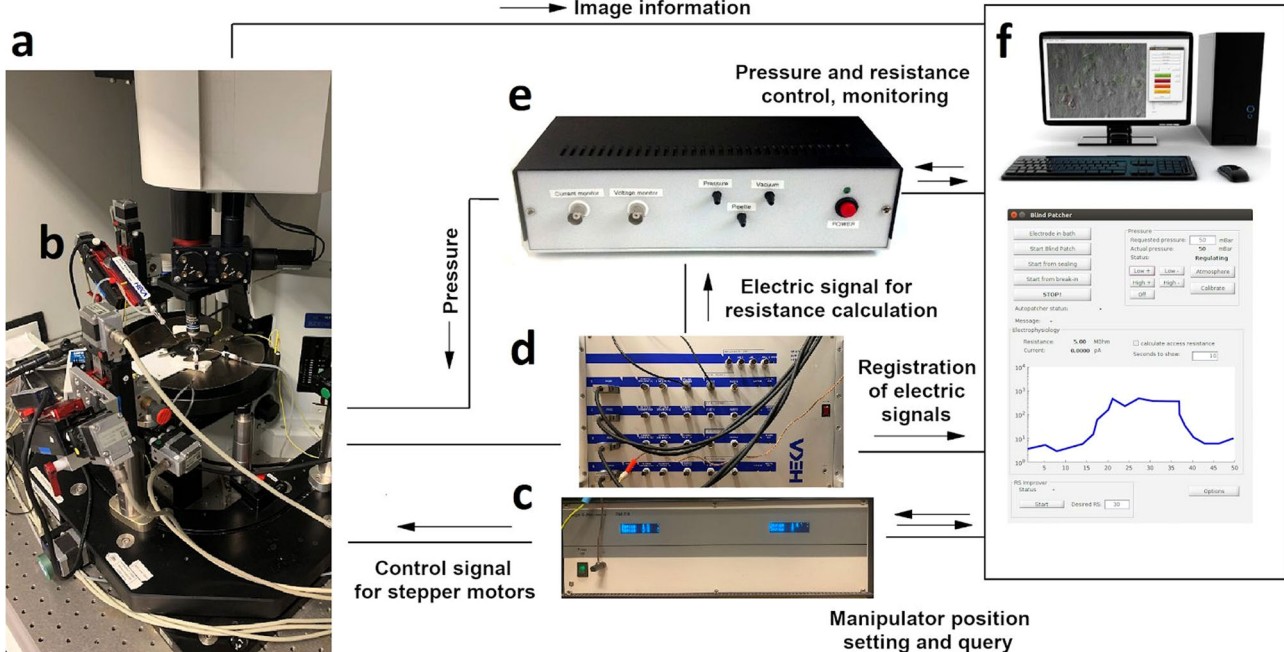

**Fig. 2 Hardware setup of the DIGAP system. a** Microscope with a motorized stage. **b** Micromanipulator. **c** Controller electronics for manipulators. **d** Patch clamp amplifier. **e** Pressure controller module. **f** Computer with the controller software.

To send commands to the amplifier, we used the "batch file control" protocol of HEKA PatchMaster 2×90.3 software (HEKA Elektronik, Germany). To apply different air pressure on the pipette in distinct phases of the patching procedure we built a custom pressure controller detailed in Supplementary Information: Pressure Regulator. Analog pressure sensors are used for monitoring the actual air pressure on the pipette and voltage signals of the sensors were connected in the input channels of the USB digitizer board. The solenoid valves of the regulator are controlled with TTL signals of the digital output channels of the digitizer.

**Pipette calibration and automatic detection**. Pipette calibration is a one-time process which determines the coordinate system transformation between the pipette and the stage axes. The calibration consists of moving the pipette along its axes with known distances, finding it with the stage and detecting the exact pipette tip position in the camera image. Calibration allows the pipette to be moved at any position of the microscope stage space. Note that no assumptions are made on the orientation or the tilt angles of the pipette.

The glass pipettes usually differ in length, thus the tip position should be updated after a pipette change. To automate this step we have developed algorithms for pipette detection in DIC images. First, we use a fast initialization heuristic and then refine the detection. The refinement step is the extension of our previous differential geometry-based method to three dimensions[21]. The pipette is modeled as two cylinders that have a common reference point and an orientation. The model is updated by the gradient descent method such that it covers dark regions introduced by the pipette in the image. Figure 3a shows the starting and final state of the algorithm from different projections in gradient images for visualization purposes. The detailed description of the algorithms and the equation derivations can be found in Supplementary Information: Pipette Detection System. The algorithm has an accuracy of $0.99 \pm 0.55 \, \mu m$ compared to manually selected tip positions, that makes it possible to reliably reach cells of $10 \, \mu m$ diameter (on average) with the pipette when oriented towards their centroids.

**Cell detection**. We applied a deep learning algorithm in order to detect cells in DIC images and propose them for automatic patch clamp recording. Various software solutions were developed to detect[25,27] or segment[28,29] neurons (and cells in general) in cell cultures or tissues, however, they do not provide satisfactory results on images of contrast-enhancing techniques such as DIC or oblique. To obtain a reliable object detection in brain tissue, we designed a cell detection algorithm, which involved three steps: data annotation, training of the model, and inference.

For acquiring an appropriate set of labeled objects, we created and included a labeling tool into the software (see Supplementary Information: Software Usage) that offers a platform to generate an annotated dataset. Field experts labeled 6344 cells on 265 stacks (184 rat, 81 human). The annotation procedure consisted of putting bounding boxes around the recognized cells over multiple slices in the stack. The stacks consisted of 60–100 slices depending on the image quality in the actual sample. The dimension of the individual slices is $1392 \times 1040$ pixels (FoV $160.08 \times 119.6 \, \mu m$). The living cells were labeled on the slices such that a 2D bounding box was put in the 3D center of each object. We also copied the same boxes to the next two slices above and below. This resulted in a bounding box that has five-slices depth. The collected labeled data was converted into the required input format of the deep learning framework we used.

We have tested four different object detection deep learning architectures, including DetectNet[30,31], Faster Region-based Convolutional Neural Network (FRCNN)[32,33], Darknet-ResNeXt[34,35], and Darknet-YOLOv3-SPP[36]. A detailed description and performance comparison is given in (Supplementary Information: Cell Detection System). DetectNet and FRCNN have been implemented into DIGAP software. The former has lower performance but very high efficiency in inference speed, while the latter is the opposite. Users can choose based on requirements and available resources. For this work we used DetectNet.

DetectNet[30,31] architecture was trained using NVIDIA's Deep Learning GPU Training System (DIGITS[37]), which is an extension of Caffe[38], and allows even the non-advanced deep learning users to perform training. The solver used for the training process was adaptive moment estimation[39] (ADAM). The pre-trained weights of the ImageNet dataset were used for the initialization of GoogLeNet to speed up the training process. The number of epochs was 2500 which took 6 days and 15 h.

FRCNN with ResNet50 backbone was also pretrained on ImageNet. The Stochastic Gradient Descent with Momentum (SGDM)[40] was used as the optimizer with cross-entropy loss function. The number of epochs was 6. The initial learning rate was 1e−3, which was dropped every 2 epochs by a factor of 0.2. The training method was set to "end-to-end", that simultaneously trains the region proposal and region classification subnetworks. MATLAB R2019b was used for training, which took 2 days and 11 h. The prediction time of a single image using DetectNet was 0.1 s, while FRCNN required approx. an order of magnitude more time, 0.96 s per image.

By using these tools, the training processes generated models that recognize neurons in their original environment in DIC images (Fig. 3b). We also implemented a procedure that extends the 2D detection by uniting overlapping bounding boxes along the Z-axis in the image stacks to complete the object detection in 3D space (Fig. 3c). Bounding boxes of different Z slices are compared and if their intersection is at least 60% of the smaller box then they are united. The following detections are compared iteratively with the intersection region. To compensate for the detection errors when cells are not detected, bounding boxes that are three slices away from each other can still be united even if the two slices in between do not contain detections.

To evaluate the performance of the proposed frameworks we measured precision, recall, and $F1$ score on a validation dataset (Fig. 3d). This dataset consisted of three image stacks (305 images in total) annotated by the same annotator and was not used in the training process. The detected objects were matched with ground truth data automatically if their centroid were at most $5 \, \mu m$ in the lateral plane and $3 \, \mu m$ in the Z axis from each other. If a detection could not be matched, it was treated as a false positive (FP). Ground truth objects not paired with a detection were treated as false negatives (FN). Based on these aspects the detection accuracy was calculated as precision $P = TP/(TP + FP)$, recall $R = TP/(TP + FN)$, and $F1$ score $= 2 * P * R/(P + R)$. DetectNet achieved 56.88% $F1$-score (precision = 53.04%, recall = 61.33%). FRCNN architecture provided better results with a 65.83% $F1$-score (precision = 60.73%, recall = 71.88%). Furthermore, the authors of the DeNeRD model[27] showed that simpler neural networks can be used to achieve good accuracy in object detection tasks. Therefore, we have compared the ResNet50 backbone to MobileNetV2[41] combined with FRCNN (Supplementary Information: Cell Detection System). This showed that MobileNetV2 can be a good compromise if hardware limitations or inference speed is an issue.

To test the performance of the annotators we have determined intraexpert and interexpert accuracies. These were measured by

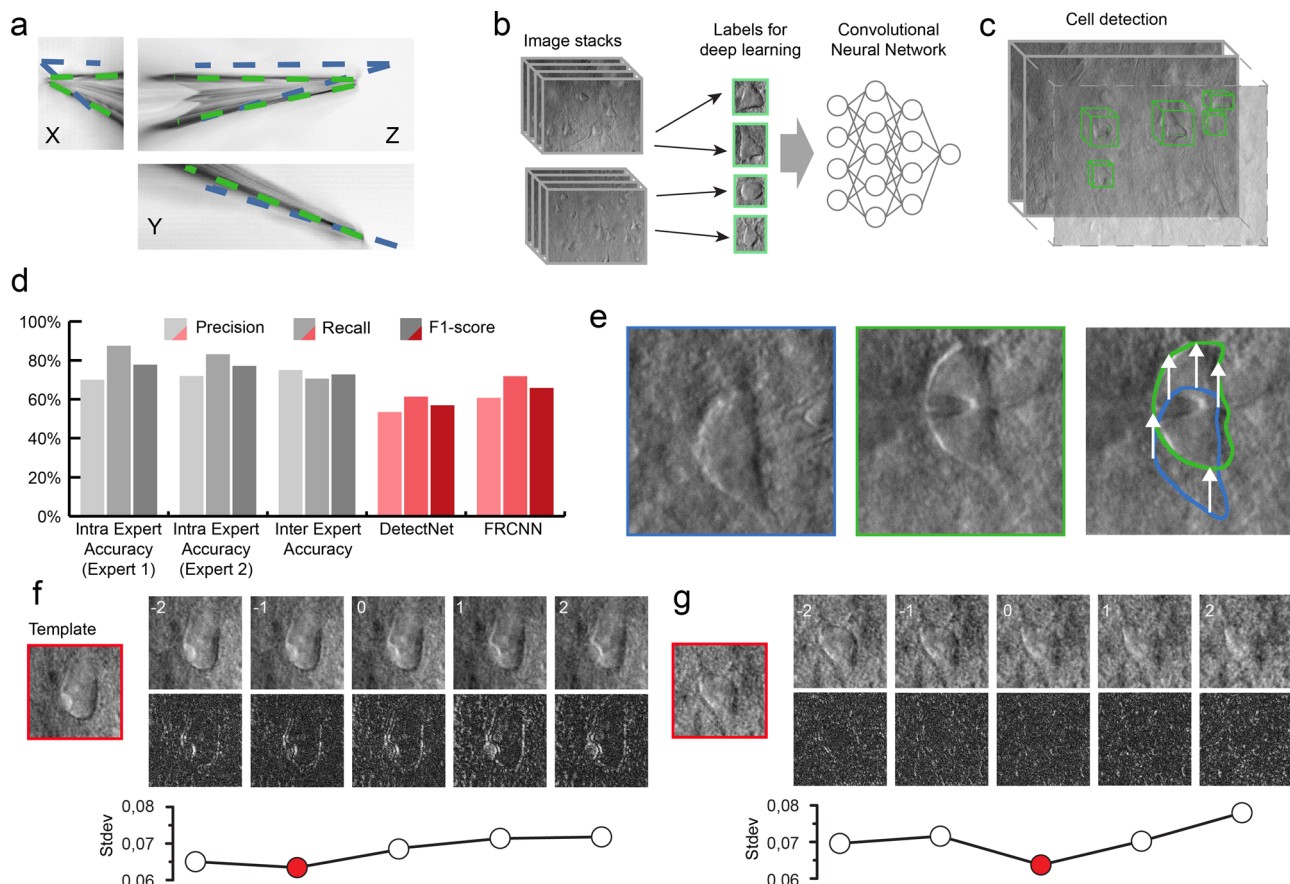

**Fig. 3 The developed algorithms for the DIGAP system. a** Result of the Pipette Hunter detection model shown in three different projections of the image stack. Initial state (blue contour) and the result (green contour) of our pipette localization algorithm are shown. **b** Training dataset generation: 265 image stacks (60–100 images per stack with 1 μm frame distance along the Z-axis) captured from human and rodent neocortical slices with DIC videomicroscopy (left). 31,720 objects as healthy cells (green boxes) labeled on every slice of the image stack by four experts. **c** After the training session, the DIGAP system detects cells in unstained living neocortical tissues. **d** Accuracy of the automated cell detection pipeline. **e** Lateral tracking of the cell movement (n = 174). DIC images of the targeted (in blue box) and patched cell (in green box). The cell drifted from its initial location (arrows in the right panel) during the pipette maneuver. **f**, **g** Z-tracking of the cell movement (n = 174). The template image was captured at the optimal focal depth (in red boxes) before starting the tracking. During the pipette movement, image stacks were captured from the targeted cell (upper panels) such that the middle slice was taken of the most recent focus position. The bottom row shows the differences between the template and the image of the corresponding Z position. The lowest standard deviation value of the difference images (plots) shows the direction of the cell drift in the Z-axis. Source Data is available as a Source Data file.

showing the same image stack (102 images) of the validation dataset to two annotators twice within 3 months time shift. The annotators reached 77.12% (precision = 71.91%, recall = 83.12%) and 77.78% F1-score (precision = 70%, recall = 87.5%), respectively. To compare the experts, the interexpert accuracy was measured which resulted in 72.73% F1-score (precision = 75%, recall = 70.59%) (Fig. 3d).

When the user initiates cell detection in the software, a stack is created and the detected cells are highlighted with bounding boxes (Fig. 3c). The detections are ordered by the confidence value, thus healthier cells are offered earlier. The target cell can also be selected manually based on arbitrary criteria required for the experiment.

**Tracking the cell in 3D**. Due to the elasticity of the tissue, the movement of the pipette can significantly deform it and change the location of the cell of interest. In order to precisely re-define the pipette trajectory, the location of the target cell needs to be tracked. We have developed an online system that performs tracking in the lateral and Z directions (Fig. 3e–g). Both directions require a template image of the target cell which is acquired

before starting the patch clamp process when the cell is in the focal plane of the microscope. The lateral tracking is performed in the image of the most recent focal level. It uses the Kanade–Lucas–Tomasi (KLT) feature tracker algorithm[42,43]. The Z tracking is based on a focus detection algorithm that operates on a small image stack encompassing the target cell body. The standard deviation of the images of the target cell body is computed and compared to initial images. As a result, the displacement direction of the target cell along the Z axis is determined. The whole process was done with stopped pipette to ensure that the cell is not pushed away meanwhile. The detailed explanation of the algorithms with examples can be found in Supplementary Information: Cell Tracking System.

**Automated patch clamping steps**. After pipette calibration and cell detection the patch clamping procedure can be started. First, the DIGAP software calculates the trajectory of the pipette movement along which the manipulator moves the pipette tip (stepwise, 2 μm) close to the cell while applying medium air pressure (50–70 mbar). The initial trajectory is a straight line along the manipulator's X axis. Note that this is tilted (in our case

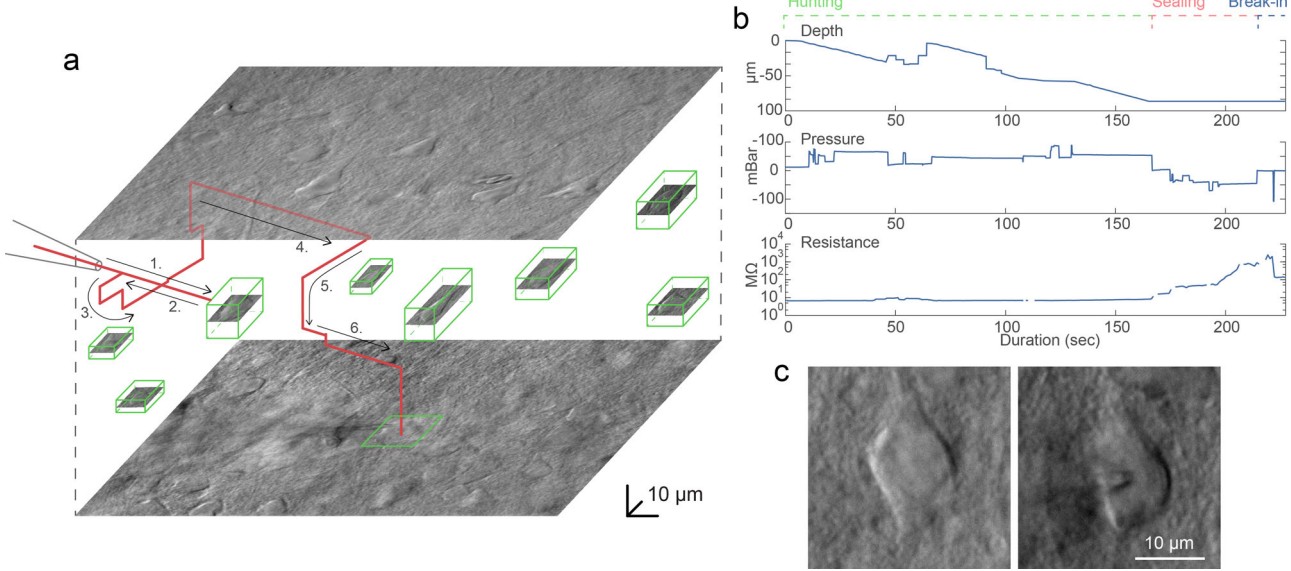

**Fig. 4 A representative example of a visual patch clamping procedure. a** Trajectory of the pipette tip (red line) with obstacle avoidance (numbered) in the tissue and the spatial location of the detected cells (green boxes). The steps of the avoidance algorithm are the following. 1: The pipette is moved forward in the initial trajectory until an obstacle is hit. 2: The pipette is pulled back. 3: The pipette is moved laterally in a spiral pattern until the resistance is back to normal. 4: The obstacle is passed. 5: The pipette is readjusted to the trajectory. 6: The approaching is continued. **b** Plots of the depth of the pipette tip in the tissue, the applied air pressure, and the measured pipette tip resistance during the approach. **c** Image of a cell before and after performing patch clamp recording on it. Source Data is available as a Source Data file.

approximately −33 degrees from the horizontal plane) so the movement vector of the pipette is parallel to the longitudinal axis of the pipette. We found that approaching is more reliable if the pipette is first moved a few micrometers above the cell and then finally descending on it. The impedance of the pipette tip is monitored continuously during the movement.

During the movement of the pipette, air pressure is dynamically changed with predefined air pressure values. Air pressures were empirically set for the different phases: hunting, sealing, and breaking. Pipette tip impedance was continuously checked in order to detect phases and apply the task-specific pressure.

Early resistance increase denotes the presence of an obstacle in front of the pipette, e.g., a blood vessel or another cell. If an obstacle is hit, the pipette is pulled back, slightly moved laterally and when the obstacle is passed the pipette is oriented back to the initial trajectory towards the target[15]. Meanwhile, the described 3D tracking algorithm compensates for the movement trajectory due to the possible displacement of the target cell. When the pipette tip reaches the target position above the cell, the pressure is decreased to a low positive value (10–30 mbar). Then the pipette is moved in the *Z* direction and the resistance of the tip is monitored by 5 ms long −5 mV voltage steps. If the impedance increases more than a predefined value (0.7–1.2 MΩ) the sealing phase is initiated. The cell-attached configuration is set up by the immediate cease of pressure. To achieve tight sealing of the cell membrane into the glass we apply small negative pressure (from −30 to −10 mbar) and the holding potential is set to −60 mV stepwise. If the sealing process is slow and does not reach 1 GΩ ("gigaseal") in 30 s, different protocols are applied. First, the initial vacuum is amplified by 1.5 and 2 times, each for 20 more sec. Then the pipette is moved +/−2 μm in each axis for 2 s. Finally, the pressure is released for 10 s and reapplied for 20 s. If the gigaseal state is reached then suction pulses (−140 to −100 mbar) of increasing length (0.5 + 0.2*attempt sec) are applied for up to 3 min to break-in the membrane. Information about the process, including pipette distance from the target,

actual air pressure, and electrical resistance values are continuously monitored and shown in the GUI windows. Description of the steps and the parameter values are described in detail in Supplementary Information: Software Usage. A representative procedure is demonstrated in Fig. 4, and further trajectory, pressure, and resistance data is visualized in Supplementary Information: Representative examples.

**Software**. The control software is written in MATLAB and the source code is made publicly available at https://bitbucket.org/biomag/autopatcher/. The visual patch clamping process can be started from a user-friendly GUI (Fig. 5) which allows every parameter to be set and the process to be monitored in real-time by the operator. Throughout the session, the Patch Clamp Diary module collects and visualizes information about patch clamping attempts, including their location and outcome status. The user can additionally mark positions in the biological sample that help orientation during the experiment (i.e., boundaries of the brain slice or the parallel strands that keep the tissue secure).

Many utility features are present to help everyday experimenting. Single images or image stacks can be acquired, saved, or loaded from the menu bar. The acquired images can be processed by performing background illumination correction or DIC image reconstruction, which can help in identifying cells and their features. The graphical processing unit (GPU) extension of our reconstruction algorithm[44] can be used for reconstruction, which results in about 1000× speed increase. The software contains a built-in labeling tool that allows image database generation to train deep learning cell recognition. Furthermore, most recent practices from other automation systems have also been implemented for the in vivo usage, including pipette cleaning[16,17] or hit reproducibility check[45]. The XML configuration file makes the adaptation easy between different setups and the software can also operate as a general microscope controller. A logging system is used for maintainability purposes.

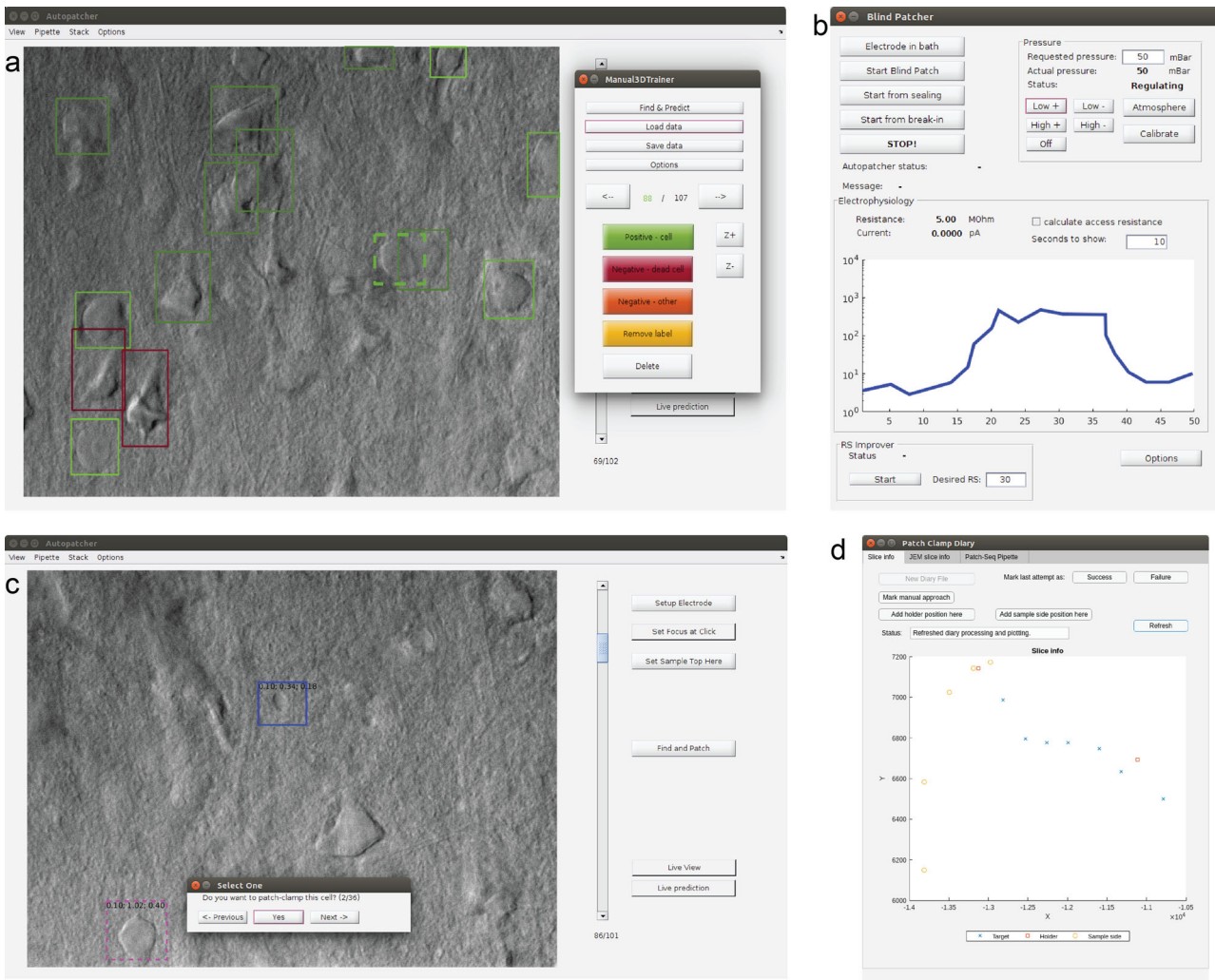

**Fig. 5 GUI of the software. a** Main window with an image stack loaded and the built-in labeling tool started. **b** Monitoring window to check the pressure and resistance values. Pressure values can be set here when operating manually, or the measurement can be restarted from different subphases here. **c** Main window when browsing the detected cells, initiated with the Find and Patch button. The measurement can be started by selecting a cell. **d** The Patch Clamp Diary module showing a plot with annotations of a sample and measurements in it.

**Application in brain slices.** To test the performance and effectiveness of our system we obtained a series of recordings (Supplementary Information: Electrophysiology stimuli for DIGAP) on slice preparation of rat somatosensory and visual cortices ($n = 23$ animals) and human temporal and association cortices ($n = 16$ patients). Successful automatic whole-cell patch clamp trials without experimenter assistance were achieved in a total number of $n = 100$ and $n = 74$ (rodent visual and somatosensory cortices and human cortex, respectively) out of $n = 157$ and $n = 198$ attempts. The data analysis was carried out using Fitmaster 2×73 (HEKA Elektronik, Germany), OriginPro 7.5 (OriginLab, USA), Excel 2016 (Microsoft, USA), and MATLAB R2017a (Mathworks, USA). The quality of recordings was supervised by measuring series resistance ($R_s$) (Fig. 6). We found a wide range of $R_s$ values within successful attempts in both species: $34.52 \pm 18.99$ MΩ in rat and $31.39 \pm 16.67$ MΩ in human recordings. Trials with $R_s$ value exceeding 100 MΩ were noted as unsuccessful attempts. Access resistance in 48.28% of our recordings was under 30 MΩ which we denoted as high quality and used for further analysis. Once the whole cell configuration was formed cells were usually held at most for 15 min to protect neuron viability for further procedures. To test the stability of whole cell configurations, we executed a separate set of experiments and found that half of the

trials ($n = 5$ out of 9) could be kept up to 1 h. The average time of experiments during the recording configuration could be maintained was $2729.9 \pm 1104.2$ s ($n = 9$, min: 928 s, max: 3825 s). During our measurements we were able to detect spontaneous postsynaptic events in the entire length of the recordings. We applied standard stimulation protocol and recorded membrane potential responses to injected currents. Based on the extracted common physiological features and firing patterns we grouped neurons into electrophysiological types (e-types[46]) based on criteria established by the Petilla convention[47]. There were eight e-types in automatic patched rat samples: pyramidal cell (pyr), burst adapting (bAD), continuous non-accommodating (cNAC), continuous stuttering (cSTUT), burst stuttering (bSTUT), delayed stuttering (dSTUT), continuous adapting (cAD), and delayed non-accommodating (dNAC). From the human samples, seven e-types were identified. In our automatically-collected dataset, dNAC type was not represented (Fig. 6).

Electrophysiological recordings were acquired using a biocytin-containing intracellular solution. We performed further anatomical investigation on $n = 44$ experiments with <30 MΩ access resistance and we achieved $n = 18$ ($n = 16$ and $n = 2$ from human and rat, respectively) full and $n = 11$ ($n = 3$ and $n = 8$ from human and rat, respectively) partial recovery

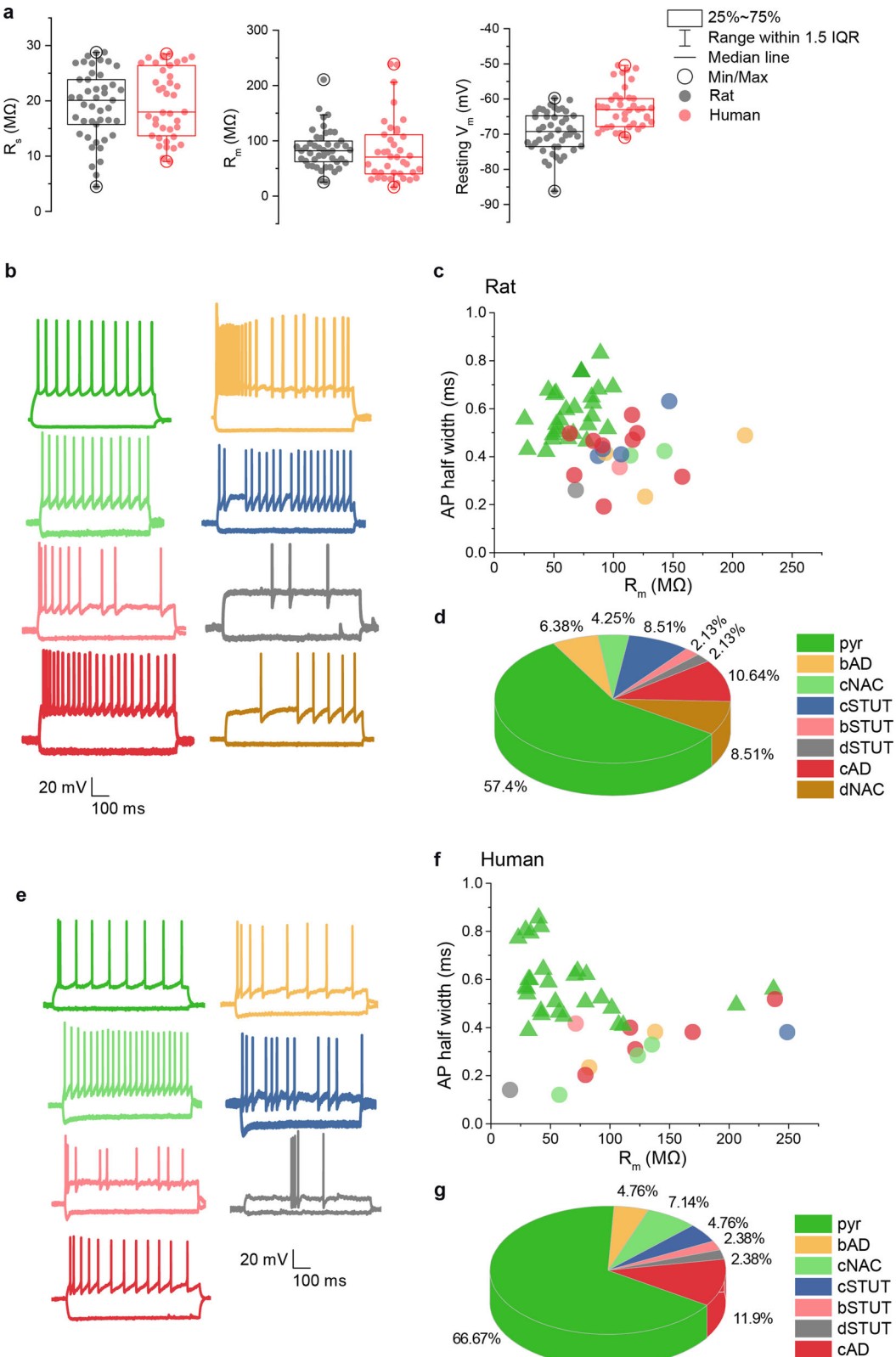

(Fig. 7a, Supplementary Information: Anatomical reconstruction examples).

We next tested if single-cell RNA analysis is achievable from the collected cytoplasm of autopatched neurons. After whole-cell recording of the neurons in the brain slices the intracellular content of the patched cells was aspirated into the recording pipette with gentle suction applied by the pressure regulator unit

(−40 mBar for 1 min, then −60 mBar for 2–3 min, and finally −40 mBar for 1 min). The tight seal was maintained and the pipette was carefully withdrawn from the cell to form an outside-out configuration. Subsequently, the content of the pipette was expelled into a low-adsorption test tube (Axygen) containing 0.5 μl SingleCellProtectTM (Avidin Ltd. Szeged, Hungary) solution in order to prevent nucleic acid degradation and to be

Fig. 6 Electrophysiological properties of the cells patched by DIGAP. a Main electrophysiological parameters from the successful automatic patch clamp recordings. The box plots show the series resistance ($R_s$, left panel), the membrane resistance ($R_m$, middle panel), and the resting membrane potential (right panel) of all successful measurements ($n = 47$ for rat and $n = 41$ for human samples). The boxes show the median, 25 and 75 percentiles, and min/max values, and the whiskers are 1.5 interquartile ranges. b Different cell types are identified according to firing features: pyr pyramidal cell, bAD burst adapting, cNAC continuous non-accommodating, cSTUT continuous stuttering, bSTUT burst stuttering, dSTUT delayed stuttering, cAD continuous adapting, dNAC delayed non-accomodating. c Individual neurons' action potential half-widths are presented as a function of the same neuron's $R_m$. Note the segregation of excitatory and inhibitory neuronal classes. Dataset is recorded from rodent samples (Panel c and d colors correspond to panel b). d The proportion of recorded cell types. e–g Same plots as b–d, representing the dataset recorded in human neocortical slices. Source Data is available as a Source Data file.

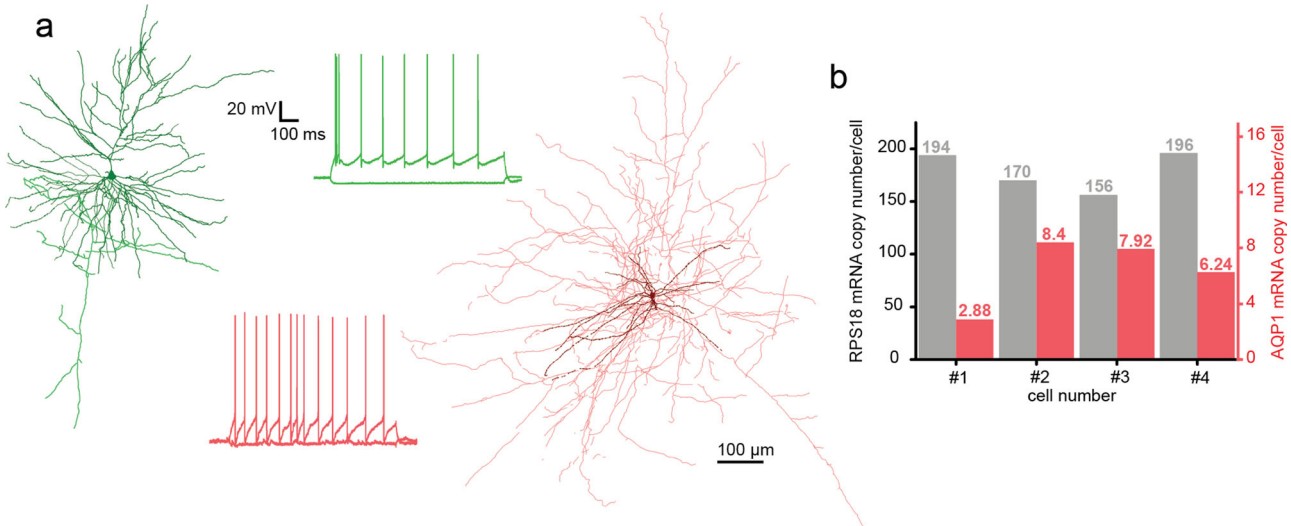

Fig. 7 Anatomical and molecular biological investigation of neurons patched by DIGAP. a Two anatomically reconstructed human autopatched neurons. The darker colors represent somata and dendrites of the pyramidal (green) and the interneuron (red) cells. The brighter color shows the axonal arborization. The firing patterns of the cells are the same color as their reconstructions. b mRNA copy numbers of a housekeeping (RPS18, black bars) and the aquaporin 1 (AQP1, red bars) gene from four representative human pyramidal cells. Source Data is available as a Source Data file.

compatible with direct reverse transcription reaction. Then the samples were used for digital polymerase chain reaction (dPCR) analysis to determine the copy number of selected genes. From four single pyramidal cell cytoplasm samples which were extracted from the human temporal cortex, we determined the copy number of a ribosomal housekeeping RPS18 and aquaporin 1 (AQP1) genes (Fig. 7b). The results of the dPCR experiments are in agreement with our previous observations[48,49].

## Discussion

The developed DIGAP system is able to fully automatically perform whole-cell patch clamp recordings on single neurons in rodent and human neocortical slices (Supplementary Movie 1, 2, 3). This is a step forward towards characterizing and understanding the phenotypic heterogeneity and cellular diversity of the brain. The presented system has a cell detection module in label-free imaging, which is achieved by deep learning. The system we developed is fully controlled by a single software, including all hardware components, data handling, and visualization. The control software has its highly comprehensive internal logging system, that allows tracking the parameters of each patch clamp recording attempt in addition with the option to store details of the cytoplasm harvesting process. In addition, it can connect to and save database entry records that are compatible with the Allen Brain Atlas single neuron database. In this work, we demonstrated the power of our system that is capable of measuring a large set of rodent and human neurons in the brain cortex. The results show strong correlation to the earlier results in literature in terms of quality and

phenotypic composition of cell heterogeneity. Records of measured cells were inserted to the database of the Allen Institute for Brain Science and a subset of the cells was isolated from their tissue environment and single-cell mRNA copy numbers of two selected genes were determined. Furthermore, we successfully demonstrated that autopatched neurons can be anatomically reconstructed.

The main advantage of the proposed system is that it can easily be integrated into any existing setups and although we do not believe that it will fully substitute human experts, it is a great choice for complex specific tasks, allows parallelization and speeds up discovery. It is important to emphasize the need for a standardized and fully documented patch clamping procedure, which is guaranteed by using DIGAP. The choice of advanced image analysis and deep learning techniques made it possible to work with the least harmful imaging modalities at a human expert level of single-cell detection that was impossible so far. Further possibilities are more widespread and potentially enabling or accelerating discoveries. Combining with intelligent single-cell selection strategies of the detected cells, the proposed system can be the ultimate tool to reveal and describe cellular heterogeneity. In multiple patch clamp setup it can be used to describe the connectome at cellular level. We presented DIGAP's application to brain research, but other fields, such as cardiovascular or organoid research will benefit from the system. Based on its nearly complete automation, it can help in education.

Future work includes adding multipipette support to study connections between pairs, triplets, or a higher number of cells at a time. Furthermore, the cell detection can be improved by increasing the size of the training dataset, the diversity of images

(by collecting them from various setups), and improving the annotation process, or even extending it to 3D instance segmentation instead of object detection.

## Methods

**Hardware setup**. A customized Olympus BX61 (Olympus, Japan) microscope with a 40× water immersion objective (0.8 NA; FoV 0.6625 mm; Olympus, Japan) with motorized Z axis (Femtonics, Hungary) which is controlled by API calls to the software was used for imaging. For moving the pipette and the microscope stage we used Luigs & Neumann Mini manipulators with SM-5 controllers (Luigs & Neumann, Germany). The electrophysiological signals were measured by a HEKA EPC-10 amplifier (HEKA Elektronik, Germany). The signals were digitized at 100 kHz and Bessel filtered at 10 kHz.

**In vitro preparation of human and rat brain slices**. All procedures were performed according to the Declaration of Helsinki with the approval of the University of Szeged Ethics Committee. Human slices were derived from materials that had to be removed to gain access for the surgical treatment of deep-brain tumors, epilepsy, or hydrocephalus from the association cortical areas with written informed consent of female ($n = 9$, aged $48.2 \pm 26.6$ years) and male ($n = 7$, aged $48.3 \pm 9.9$ years) patients prior to surgery. Anesthesia was induced with intravenous midazolam and fentanyl (0.03 mg/kg, 1–2 μg/kg, respectively). A bolus dose of propofol (1–2 mg/kg) was administered intravenously. To facilitate endotracheal intubation, the patient received 0.5 mg/kg rocuronium. After 120 s, the trachea was intubated and the patient was ventilated with a mixture of $O_2$ and $N_2O$ at a ratio of 1:2. Anesthesia was maintained with sevoflurane at monitored anesthesia care (MAC) volume of 1.2–1.5. After surgical removing blocks of tissue were immediately immersed in ice-cold solution containing (in mM) 130 NaCl, 3.5 KCl, 1 NaH$_2$PO$_4$, 24 NaHCO$_3$, 1 CaCl$_2$, 3 MgSO$_4$, 10 d(+)-glucose, saturated with 95% O$_2$ and 5% CO$_2$. Slices were cut perpendicular to cortical layers at a thickness of 350 μm with a vibrating blade microtome (Microm HM 650 V, Thermo Fisher Scientific, Germany) and were incubated at room temperature for 1 h in the same solution. The artificial cerebrospinal fluid (aCSF) used during recordings was similar to the slicing solution, but it contained 3 mM CaCl and 1.5 mM MgSO$_4$.

Coronal slices (350 μm) were prepared from the somatosensory cortex of male Wistar rats (P18-25, $n = 23$, RRID: RGD_2312511)[50]. All procedures were performed with the approval of the University of Szeged and in accordance with the Guide for the Care and Use of Laboratory Animals (2011). Recordings were performed at 36 °C temperature. Micropipettes (3.5–5 MΩ) were filled with low [Cl] intracellular solution for whole-cell patch clamp recording: (in mM) 126 K-gluconate, 4 KCl, 4 ATP-Mg, 0.3 GTP-Na$_2$, 10 HEPES, 10 phosphocreatine, and 8 biocytin (pH 7.20; 300 mOsm).

**Molecular biological analysis**. After harvesting the cytoplasm of the recorded cells the samples were frozen in dry ice and stored at −80 °C until used for reverse transcription. The reverse transcription (RT) of the harvested cytoplasm was carried out in two steps. The first step took 5 min at 65 °C in a total reaction volume of 5 μl containing 2 μl intracellular solution and SingleCellProtectTM mix with the cytoplasmic contents of the neuron, 0.3 μl TaqMan Assays, 0.3 μl 10 mM dNTPs, 1 μl 5× first-strand buffer, 0.3 μl 0.1 mol/l DTT, 0.3 μl RNase inhibitor (Life Technologies, Thermo Fisher Scientific, Germany) and 100 U of reverse transcriptase (Superscript III, Invitrogen, Thermo Fisher Scientific, Germany). The second step of the reaction was carried out at 55 °C for 1 h and then the reaction was stopped by heating at 75 °C for 15 min. The reverse transcription reaction mix was stored at −20 °C until PCR amplification. For digital PCR analysis the reverse transcription reaction mixture (5 μl), 2 μl TaqMan Assays (Life Technologies, Thermo Fisher Scientific, Germany), 10 μl OpenArray Digital PCR Master Mix (Life Technologies, Thermo Fisher Scientific, Germany) and nuclease-free water (5.5 μl) were mixed in a total volume of 20 μl. The mixture was evenly distributed on an OpenArray plate. RT mixes were loaded into four wells of a 384-well plate from which the OpenArray autoloader transferred the cDNA master mix by capillary action into 256 nanocapillary holes (four subarrays) on an OpenArray plate. Processing of the OpenArray slide, cycling in the OpenArray NT cycler and data analysis was done as previously described[48]. For our dPCR protocol amplification, reactions with CT confidence values below 100 as well as reactions having CT values less than 23 or greater than 33 were considered primer dimers or background signals, respectively, and were excluded from the data set.

**Anatomical processing and reconstruction of recorded cells**. Following electrophysiological recordings, slices were transferred into a fixative solution containing 4% paraformaldehyde, 15% (v/v) saturated picric acid, and 1.25% glutaraldehyde in 0.1 M phosphate buffer (PB; pH = 7.4) at 4 °C for at least 12 h. After several washes with 0.1 M PB, slices were frozen in liquid nitrogen then thawed in 0.1 M PB, embedded in 10% gelatin, and further sectioned into 60-μm slices. Sections were incubated in a solution of conjugated avidin-biotin horseradish peroxidase (ABC; 1:100; Vector Labs) in Tris-buffered saline (TBS, pH = 7.4) at 4 °C overnight. The enzyme reaction was revealed by 3′ 3-diaminobenzidine tetrahydrochloride (0.05%) as chromogen and 0.01% H$_2$O$_2$ as oxidant. Sections were

postfixed with 1% OsO$_4$ in 0.1 M PB. After several washes in distilled water, sections were stained in 1% uranyl acetate and dehydrated in an ascending series of ethanol. Sections were infiltrated with epoxy resin (Durcupan) overnight and embedded on glass slides. Three-dimensional light-microscopic reconstructions were carried out using a Neurolucida system (MicroBrightField, USA) with a 100× objective.

**Pipette cleaner**. We implemented a pipette cleaning method[16] into our system. The cleaning procedure requires two cleaning agents: Alconox, a commercially available cleaning detergent, and artificial cerebrospinal fluid (aCSF). We 3D printed a holder for two PCR tubes containing the liquids that can be attached to the microscope objective and are reachable by the pipette tip. The cleaning is performed by pneumatically taking up and then removing the agents into and from the pipette. The vacuum strength used for the intake of the liquids is −300 mBar and the pressure used for the expulsion is +1000 mBar. The method consists of three steps. First, the pipette is moved to the cleaning agent bath and vacuum is applied for 4 s. Then, to physically agitate glass-adhered tissue, pressure and vacuum are alternated, each for 1 s and repeated for five times total. Finally, pressure is applied for 10 s to make sure all detergent is removed. In the second step, the pipette is moved to the aCSF bath and any remaining detergent is expelled by applying pressure for 10 s. In the third step, the pipette is moved back to the position near to the biological sample where the cleaning process was initiated. In the original paper, it is shown that these pressure values and the duration of the different steps are more than enough to cycle the volume of agents necessary to clean the pipette tip. We provide a graphical window in our software to calibrate the pipette positions of the tubes containing the cleaning agent and the aCSF and to start the cleaning process.

**Reporting summary**. Further information on research design is available in the Nature Research Reporting Summary linked to this article.

## Data availability

The data that support the findings of this study are available in the manuscript, Source Data file, supplementary information and available from the authors upon reasonable request. The annotated image data used for deep learning are available from the corresponding author upon request. Source data are provided with this paper.

## Code availability

Source code is available from Bitbucket at https://bitbucket.org/biomag/autopatcher/.

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

## Acknowledgements

We thank Tímea Tóth and Réka Hollandi for their help in the image labeling, Ádám Szűcs for his work in the early stages of the development, Tamás Szépe for the advice on manipulator control, Nelli Tóth for the anatomical reconstruction, and István Grexa for the 3D printing. This work was supported by NAP-B brain research grant; the NVidia GPU Grant program; the LENDULET-BIOMAG Grant (2018-342); the European Regional Development Funds (GINOP-2.3.2-15-2016-00006, GINOP-2.3.2-15-2016-00026, GINOP-2.3.2-15-2016-00037); the Loránd Eötvös Research Network; the National Research, Development and Innovation Office of Hungary (GINOP-2.3.2-15-2016-00018, KKP_20 Élvonal KKP133807) (G.T.); Ministry of Human Capacities Hungary (20391-3/2018/FEKUSTRAT) (G.T.); from the National Research, Development and Innovation Office (OTKA K128863) (G.T., G.M.); ÚNKP-20-5 - SZTE-681 New National Excellence Program of the Ministry for Innovation and Technology from the source of the National Research, Development and Innovation Fund (G.M.); and János Bolyai Research Scholarship of the Hungarian Academy of Sciences (G.M.).

## Author contributions

K.K. developed the software. G.O., K.K., M.R., and A.O. built and assembled the hardware. K.K. performed imaging. T.B. and K.K. developed the cell detection system. A.O., G.M., G.O., K.K., N.M., and M.R. performed electrophysiological measurements and analyzed the data. E.T. and J.M. developed the reconstruction models. P.B. provided human samples. G.M., G.T., and P.H. supervised the project. K.K, G.O., T.B., N.M., M.R., A.O., J.M., G.M., G.T., and P.H. contributed to the manuscript.

## Competing interests

The authors declare no competing interests.
