## [Peer Review File · Nature Communications]

Reviewers' Comments:

Reviewer #1:

Remarks to the Author:

There have been substantial advances in automated patch-clamp technology for intact tissue preparations in recent years. This paper describes a successful attempt to produce a near-fully automated in vitro patch clamp system, with an aim of increasing the throughput of electrophysiological characterisation in label-free tissue slices. The near-full automation could be a great advantage in this regard, allowing one operator to control multiple rigs, or allowing an operator to perform patch-clamp recordings without the extensive periods of training currently required. However, it is not quite the first such system, and there do not appear to be substantive advances over the work by Ilya Kolb (2019 J Neural Engineering).

Although the paper describes an impressive body of work, I have many minor issues with it:

- Line numbering in the manuscript provided would have aided the provision of reviewer feedback
- In several points in the manuscript, it is mentioned that "the nucleus or cytoplasm can be harvested" (step 10 in introduction). It is not clear if this is a part of the operation of the automated system, and how this is performed is not described in the paper. (if it is, this could have been emphasized, as I am not aware of another automated system with this capability). This must be clarified – if the user has an option at this stage to do this manually, this is fine, but the paper should be clear on the point. If it is automated, it must be described.
- Some details of the deep learning algorithm used should be given in the Methods section in the main paper, not just left to the supplementary information. The paper must stand on its own, and adequate methodological information is not present without at least a brief summary of this. The methodological information provided in the supplementary information is also inadequate. Essentially, it seems as if the authors used the Caffe framework as a black box. Much more information on the
- Proper formatting of the supplemental material (such as paragraph indenting) should be carried out. (This also applies to the main text, however that at least will get the benefit of journal typesetting). What precisely was the model architecture, what hyperparameters were chosen, and why? What are the characteristics of the training data (pixels, dimensions etc). How long did the algorithm take to train, and to classify?
- The system works on the basis of a training dataset collected by 4 experts who labelled "healthy cells". However, the "inter-expert" accuracy seems to be low, suggesting that the quality of the training dataset may not be high. The claim that the deep learning model is outperforming the annotators is unjustified – without ground truth data, all that can be said is that each annotator and the algorithm had different performance. I would recommend applying the system to labelled data (in addition to label-free) in order to use fluorescence measurements to obtain performance measures.
- No information is given on the DIC optics used to acquire the data processed. What objective lens is used, what is the field of view?
- What proportion of data is from human and what from rodent data? Does this bias the results in any way? If the system is trained on just the rodent, does it generalise to the human data? Is the rodent data from mouse or rat?
- In Fig. 2, why are only 2 of the 4 experts shown?
- The performance analysis needs to be spelled out in more detail. To obtain true positives, false positives etc, the ground truth needs to be known. What *exactly* was the "ground truth" used for the precision and recall calculations? Is it the superset of the expert annotations? Did no experts cover the same dataset in the initial dataset? I am still unclear on this, despite reading the extended description in the Supp Material.
- Fig. 2f-g: The purpose of this panel is to apparently show the drift in the cell as the pipette is lowered into the tissue. However, the figure needs a lot more clarification. What is the reasoning behind the template not being taken at the 0th image position and what is the numbering exactly if not an image position with respect to the template? Is it with respect to just the middle image?

Then the standard deviation is taken for the entire difference image, which then can be used to show the drift in z-axis. The plots don't clearly indicate that it is showing the standard deviation which should be shown. There are no error bars on this either. In the right-hand plot, the standard deviation for the position of the template image is comparable to the -2 and -1 positions on the left plot. Would be nice to show a control plot where there is not pipette movement and image stacks are taken repeatedly and compared to the same template image. This would show if the standard deviation of the difference images really show anything at all.

- "A central part of the method is the detection of single neurons in label-free 3D images using deep convolutional neural networks reaching super-human precision.". This is a bold claim, considering that the precision of experts 1 and 2 in Figure 2d are higher than the precision of the detection algorithm. No evidence of super-human precision is demonstrated, and to do so would require eg a fluorescent label for ground truth, ie. It is not something that even could be demonstrated using the approach that has been taken – leaving aside that the claim seems to be false according to the authors' own presentation.
- Regarding the trajectory, what angle is taken relative to the horizontal? This is an important piece of data.
- Why is recording quality measured only by R_s, rather than using R_{in}, which is more common? Are the 4 cells with R_{in} above 200 MOhms really acceptable?
- Fig 4b should also have scale bars (not just depending upon looking down to 4e)
- Fig 4 panels c and d are swapped relative to the caption
- Penultimate paragraph, "integrated into" would work better than "portable to".
- Please use either indentation or paragraph spacing to separate paragraphs, it is vey difficult to read otherwise.
- The final paragraph begins with a rather ambitious aspiration. I do not believe that there is sufficient information in the DIC image to do this.

Reviewer #2:

Remarks to the Author:

General Remarks

The study by Koos et al. (Automatic deep learning driven label-free image guided patch clamp system for human and rodent in vitro slice physiology) presents a hardware and software toolbox for automated patch-clamp recordings in in vitro acute slice preparations from rodent and human brain. The authors use deep learning for cell detection and a subsequent pipeline for pipette localization, tracking and ultimate accessing the neuron through whole-cell configuration. This manuscript, is part of a series of efforts in the neuroscience field for automating the laborious process of recording the electrophysiological properties of cells, intrinsic and synaptic, as well as obtaining the morphological characteristics of the recorded neuron after filling it with a dye (typically biocytin or neurobiotin). In this case the authors also added mRNA profiling by aspirating the cell nucleus and cytoplasm after recording to detect the expression of key genes using digital PCR. Overall, the efforts presented in the manuscript are timely and aim at optimizing and streamlining a series of processes to present an end-end solution for performing high-throughput electrophysiological and molecular analysis of single cells in unstained tissue samples. Unfortunately, despite the great lengths that the researchers have gone to for this work, the study has the drawback that both the usage of deep learning for cell detection and the automation of patch-clamping have been implemented before independently, also in publications some of which the authors are citing (for ex. Kodandaramaiah et al Nat. Protocols 2016; Wu Q et al. J. Neurophysiology 2016; Suk SJ et al. Neuron 2017 for automated patching and Ounkomol C et al. Nature methods 2018; YoungJu Jo et al. arxiv 2018 for label-free cell and intracellular organelle detection). Nevertheless, the combination of methods that the authors have put together, from the usage of AI to detect unlabeled cells on brain slices and subsequently, not only

electrophysiologically record these cells, but also reveal their morphology and some of their genetic makeup has not been shown before.

Major issues:

Regarding the Machine Learning part:

- Although a big focus of the study (as indicated in the title as well) is targeted in using deep learning for cell detection, the study unfortunately lacks a survey of existing cell detection techniques developed through deep learning-based methods. A quantitative comparison should be provided with other deep learning-based methods to show that the algorithm the authors have used performs better than other cell detection techniques (e.g. Iqbal et. al. Sci Reports. 2019, Zhou, Zhi, et al. Brain informatics 2018, Falk, Thorsten, et al. Nature Methods 2019), if trained on the same dataset.

- Also, it would help the reader if a detailed explanation of the deep neural network architecture is included, and the same goes for an explanation of why DetectNet would perform well on unlabeled bright field images of neurons, as compared to other deep learning and non-deep learning-based cell detection tools (e.g. Iqbal et. al. Sci Reports. 2019, Zhou, Zhi, et al. Brain Informatics 2018, Falk, Thorsten, et al. Nature Methods 2019).

- Even though based on the accuracy score presented in Figure 2d, the algorithm seems to outperform human experts, the reviewer is of the opinion that neuron detection/localization (bounding box) is not optimal to get a 3D representation of the cell body for patching; if the neuron was segmented in 3D (also possible in 2D then digitally turned into 3D) a higher accuracy would probably be achieved (e.g. Januszewski, Michał, et al. Nature Methods 2018).

- It is not clear from the manuscript how much (what is the ratio) of the human annotated dataset was used for training and validation? Was there any cross-validation performed? A precision-recall curve is missing, as is a quantitative plot that shows the error rate of the network while training. It is not clear what kind of neuron images were used for training and testing, potentially the majority of the training and testing data is drawn from the same set hence the F1 score seems higher, which could simply be the result of overfitting.

- According to the authors, pre-trained weights of ImageNet were used, so I suppose transfer learning was applied. How many layers in the network were frozen during training? It would be good if the authors provided an explanation why GoogLeNet weights trained on ImageNet would work well for detecting neurons, given that ImageNet only contains natural images. Do images of unlabeled neurons in a slice have same features (statistics) as natural images (e.g. person)?

Regarding the electrophysiology/anatomy/mRNA part:

- When the nucleus is harvested, the morphology of the cell usually cannot be maintained. What was the success rate for obtaining both?

- An access resistance of 30MΩ is high for whole-cell patch-clamp recordings. It may be that the intrinsic electrophysiological properties of the cells are not that affected by this high R_s , but the synaptic events would. Have the authors tried to record synaptic events using their system? Can the system change between current and voltage clamp automatically?

- How stable were the recordings? The authors have a log system which registers all the values at any given point. How much did the access resistance change over time and how much time could the authors keep the cells stable and healthy?

- In figure 5:

It would be good if the authors showed more examples of the reconstructed morphology of the recorded cells or at least images of z-projections in the supplementary figures. In this reviewer's experience a success rate of 80% in the recovery of morphology is very high indeed. This is even more surprising given that the authors also suck up the nucleus of the cell for RNA analysis.

In the Farago N et al. paper that the authors reference, the digital PCR method was used to quantify many genes that have specific biological functions, such as the delta subunit of GABA_A receptors, *slc2a4* and microRNAs. Here the authors show the expression level of only two genes, which encode for proteins that are quite generic. It may be that these genes would have also been detected if aspiration of the extracellular debris was occurring instead of the cell nucleus. Have the authors performed a control experiment where they purposefully aspirate extracellular material to compare with the results they get in Figure 5b? The inclusion some more specific genes would be help assess how well this automated part works.

- Finally, it would be nice if the authors presented a zoomed out video clip that would showcase how this system looks like, including the microscope, amplifier, automated manipulator etc, as well how it works once a slice is put down the chamber.

Minor Issues:

There is no line numbering provided so as to include it herein, but the text where changes may be needed is provided below in italics.

- Please consider rephrasing the sentence: The quantitative and qualitative efficiency of single-cell patch clamp procedure is highly determinant for every follow up measurements including anatomical reconstruction and molecular analyses.

- Please change: Recently, patch clamp technique has to Recently, the patch clamp technique has

- Regarding the sentence: Blind patch clamping' moves the pipette forward in vivo
Blind patch clamping was first done in vitro and only later performed in vivo

- Please delete "of" in the sentence: electrophysiological measurements strongly correlates to that of made by a trained

- It is stated that the arrows in Figure 2e are yellow, but they are white in color

- Please change the word "were" to "was" and "vacuum" to "suction" in the following sentence: intracellular content of the patched cells were aspirated into the recording pipette with gentle vacuum applied by the pressure regulator unit (-40 mBar for 1 min, then -60 mBar for 2-3min, and finally -40 mBar for 1 min).

Reviewer #3:

Remarks to the Author:

The article is focused on using deep learning for recognition of DIC images of label-free neurons in brain slices for further automatic patch-clamp. Patch-clamp is the main electrophysiological technique for single-cell recordings. It is the primary methodology for the analysis of electrical properties of neurons and other electrically excitable cells. Several recent publications from different laboratories have described automated patch clamp systems, both blind and image-guided in slices and in vivo. These different systems are able to use computer vision libraries to detect pipette tips, fluorescently labeled cells, adjust their patch path, wash and reuse the

pipettes, patch multiple cells at the same time.

The key innovation of the authors' paper is the use of deep learning first to train their neural network and then recognize the images of neurons in brain slices acquired using DIC optics. This cell recognition is then combined with an automatic patch-clamp. The software is written in Matlab with external deep learning library calls. Overall, the use of deep learning to recognize cells in DIC optics is important for the field, even though all other parts of the process have been published previously. However, there are several problems that need to be addressed, including a poor description of the deep learning part of the paper. I am enthusiastic about this paper, provided the issues below are addressed, and the manuscript is revised.

There are several issues that need to be addressed:

- 1) First, it is not clear if the patching process is fully automatic or if the interference of a human may be required, and if it is, how often. The corresponding statistics are not sufficiently extensive; only the success examples vs. failures are mentioned. What is the percentage of fully automatic vs. attempts with human interruption vs failures to patch?
- 2) The hardware configuration in Figure 1 doesn't actually show any meaningful hardware, for example, which patch-clamp amplifier, how it's connected to the National Instruments board. What kind of signals the NI board receives, and what commands does it send and where?
- 3) How do the authors control the amplifier and send/receive information? Is the output impedance then received by the NI board?
- 4) The detailed information and which commands are used by the authors to switch between voltage/current clamp modes, measure impedance, inject voltage command needs to be documented in a supplementary file in addition to the bitbucket or github or another online open-source repository.
- 5) How the z-focus is changed is not clear. This information is also important, especially in consideration of wider adoption in the neuroscience community. Different slice rigs may have different methods to change the z-focus. How do the authors control the z-focus from the software, and how does it integrate with both pipette detection, cell detection, z-stack acquisitions? A more detailed description should be both in the main and supplementary figures.
- 6) The graphical user interface(GUI) of the software is shown only in the supplemental file. There should be at least a separate main figure showing the main elements of GUI with the detailed annotation. This figure should also include the GUI for initiating the cell recognition process.
- 7) In the video shown, one can see the patching process, but not the deep learning-based recognition of the cells with further successful targeting. It would be useful to include a video of the cell recognition process.
- 8) In the video of the patching process, one can see the changes in resistance, but not the pressure. It is consequently not clear when the negative pressure was applied to break-in. How much suction was applied, etc. Is this how the process usually works? Or is there a separate pressure window which was not shown in the video? Probably a better video could be used, showing the average patching process.
- 9) There is no diagram of the cell recognition->automatic patching algorithm. Consequently, it's not clear what happens after what. At least, I was not able to easily find the algorithm diagram.
- 10) The Figure 1a panel represents a series of confusing icons that can hardly be seen and are not an informative algorithm diagram.
- 11) The paragraph describing cell recognition and specifically deep learning, including training, is very brief and needs to be vastly expanded. There should be more information, more details about how the cells are recognized, what happens to the recognized cells in each z-stack cross-section, how they are then combined, etc. This is probably the biggest innovation of the paper, yet there is almost no information about how this is done.
- 12) Figure 2 describing the algorithms, and the cell detection module is incomplete. Specifically, Figure 2b has the model mentioned, but which model? Is it a convolutional network or maybe LSTM? This is not clear, yet this information is critical and needs to be obvious from arguably the most important figure in the paper.
- 13) The information about calling Caffe libraries from Matlab is only available in the supplemental information, with only a brief mention of the requirement of a separate computer with GPUs for training vs. recognition. This kind of information about the core innovation of the paper needs to

be in the main text.

14) A lot of other information about the setup is very hard to find. For example, the authors use the 40x water immersion lens. This needs to be in the first figure describing the hardware setup. What happens if one uses a 63x lens, will it work with the software?

15) How does the system operate when you switch between a lower magnification lens (10x?) used for the initial targeting of the brain slice. Do the authors even use one? It is not clear from the article. Is the initial calibration of the pipette and the derivation of the slice->pipette transformation function necessary at the low magnification? Is it done manually or fully automatically?

16) There are several questions related to pipette detection. First, the authors cite only their own previous conference paper [21] while ignoring previous earlier work from other groups demonstrating successful automatic pipette tip detection.

17) Another confusion is related to the pipette tip detection. It is mentioned that the mouse clicking on the tip of the pipette is required for calibration. If this is correct, this may not represent the fully automatic detection of the pipette tip.

18) Figure 3a is also confusing. The path of the pipette seems very strange for an algorithm. I assume the idea here is to demonstrate a patch attempt with an obstacle hit. There is no annotation of the stages 1,2,3 in Fig. 3a. I assume 3 represents a complicated avoidance path, but this is a very complicated path. Is this performed by a human to re-position the pipette? It looks from the schematic as if some of the path includes lateral movements within the slice? Not sure.

19) Fig. 3b. There are several non-regular changes in the pipette pressure during the approach, which are strange? Were there any commands performed by the algorithm to apply positive/neutral pressure? Also, after the start of the sealing process, again, several step-like changes in the negative pressure. Again, without the algorithm or a description of their state-system, it is not clear what is going on here.

20) There need to be more representative examples of successful and unsuccessful patches (both path and diary files with resistance and pressure measurements).

Minor issues:

1) Some of the citations are not precise. For example, " ... Object detection of neurons in label-free tissue images is challenging [24]..." 24 represents a paper by the authors, which performs image recognition on the astrocytes stained using immunohistochemistry (IHC). It would probably be more appropriate to say something like: ... we have previously demonstrated the identification of labeled cells using deep learning following immunohistochemistry.

2) There needs to be more extensive discussion how different hardware can be used with the authors' DIGAP system. Also, there needs to be more discussion of how different deep learning frameworks can be potentially used with the software.

Review Answers

Our response to the comments from the reviewers

Below, we provide a detailed summary of the comments made by the reviewers. Each comment is followed by our response (*denoted in italic font*). We have responded to every comment [Rx.y] from the reviewers and made a genuine effort to address all concerns [Ax.y]. In order to clearly denote where changes have been made to the manuscript, we have **highlighted changes from the previous version of the manuscript in red**.

REVIEWER COMMENTS

Reviewer #1 (Remarks to the Author):

[R1.1] There have been substantial advances in automated patch-clamp technology for intact tissue preparations in recent years. This paper describes a successful attempt to produce a near-fully automated in vitro patch clamp system, with an aim of increasing the throughput of electrophysiological characterisation in label-free tissue slices. The near-full automation could be a great advantage in this regard, allowing one operator to control multiple rigs, or allowing an operator to perform patch-clamp recordings without the extensive periods of training currently required. However, it is not quite the first such system, and there do not appear to be substantive advances over the work by Ilya Kolb (2019 J Neural Engineering).

[A1.1] We are grateful to this Reviewer for the detailed and constructive review of our work and acknowledging it. We hope that our detailed discussions below and changes in the manuscript will answer all issues raised.

Although the paper describes an impressive body of work, I have many minor issues with it:

[R1.2] Line numbering in the manuscript provided would have aided the provision of reviewer feedback

[A1.2] We have included line numbering in the updated manuscript to assist the further review process.

[R1.3] In several points in the manuscript, it is mentioned that “the nucleus or cytoplasm can be harvested” (step 10 in introduction). It is not clear if this is a part of the operation of the automated system, and how this is performed is not described in the paper. (if it is, this could have been emphasized, as I am not aware of another automated system with this capability). This must be clarified – if the user has an option at this stage to do this manually, this is fine, but the paper should be clear on the point. If it is automated, it must be described.

[A1.3] We thank the reviewer for pointing out the lack of these details. The harvesting and anatomical reconstruction steps are manual. We have started developing a module for automated harvesting but we did not find it reliable and did not include it in the manuscript. We have clarified the text in Introduction.

[R1.4] Some details of the deep learning algorithm used should be given in the Methods section in the main paper, not just left to the supplementary information. The paper must stand on its own, and adequate methodological information is not present without at least a brief summary of this. The methodological information provided in the supplementary information is also inadequate. Essentially, it seems as if the authors used the Caffe framework as a black box. Much more information on the

[A1.4] We agree with the reviewer that important information should be present in the main text and now we have included a more detailed description of the deep learning algorithms in the main text. In addition, now we present further networks and we provide details of the settings in the supplementary document “Deep Learning Model Comparison for Cell Detection”.

[R1.5] Proper formatting of the supplemental material (such as paragraph indenting) should be carried out. (This also applies to the main text, however that at least will get the benefit of journal typesetting).

[A1.5] We applied formatting on the main text and the supplementaries for better readability.

[R1.6] What precisely was the model architecture, what hyperparameters were chosen, and why? What are the characteristics of the training data (pixels, dimensions etc). How long did the algorithm take to train, and to classify?

[A1.6] We have extended the description of the architecture, hyperparameters, time consumption, and the data characteristics that were previously in the supplementary, and moved a significant part of it to the main text. Now all the requested information can be found in the Results, Cell detection system section. Currently, the supplementary contains a comparison of 4 different cell detection approaches trained on the dataset.

[R1.7] The system works on the basis of a training dataset collected by 4 experts who labelled “healthy cells”. However, the “inter-expert” accuracy seems to be low, suggesting that the quality of the training dataset may not be high. The claim that the deep learning model is outperforming the annotators is unjustified – without ground truth data, all that can be said is that each annotator and the algorithm had different performance. I would recommend applying the system to labelled data (in addition to label-free) in order to use fluorescence measurements to obtain performance measures.

[A1.7] We thank the reviewer for pointing out this issue with the dataset and our claim. We are unsure what the Reviewer meant by labelled data here. If he/she refers to applying the system in a fluorescent environment, it is very straightforward doing so - from an image analysis point of view - on the other hand there have been several papers published in that topic and it is out of the scope of our research interest.

If this Reviewer meant using labelling in combination with DIC. We started the annotation procedure combining DIC images with live fluorescent markers, with little success. Post labeling was possible but the structural changes of the soft tissue resulted in a highly non-linear registration problem that we were unable to resolve, therefore we have decided to annotate DIC stacks. We completely agree that the quality of the training dataset might not be perfect, however, by annotating the dataset with experts we tried our best to do so. Despite this, the annotations are subjective, DIC images are difficult and cells that are dimmed and don't have sharp contours may be missed sometimes.

The deep learning model having higher performance than the annotators made us also realize that it could be an issue with the validation set we used earlier. We have therefore extended the validation dataset with further images, inspired by this comment we validated this set by a patch clamp expert, (re)evaluated the models and updated the conclusions in the text. The claim that the machine outperformed human experts was invalidated and removed from the text.

[R1.8] No information is given on the DIC optics used to acquire the data processed. What objective lens is used, what is the field of view?

[A1.8] A 40x water immersion objective (0.8 NA; Olympus, Japan) was used with a 0.6625 mm theoretical field of view. With this objective, our system created a 160.08 x 119x6 um field of view on the screen. We have updated the Methods, Hardware setup section with this information.

[R1.9] What proportion of data is from human and what from rodent data? Does this bias the results in any way? If the system is trained on just the rodent, does it generalise to the human data? Is the rodent data from mouse or rat?

[A1.9] We have created annotations for 184 rodent and 81 human image stacks (60-100 slices per each, depending on quality), which resulted in 3481 and 2542 annotated objects, respectively, on 7282 and 4928 2D images. The human data was generated later and by that time we were already experimenting on human images with the model trained on rodent data. We were satisfied with the detection quality in general, however, we noticed that some human-specific, large neurons were not detected. The model trained on the combined dataset helped with this issue. The rodent dataset contains rat images only. Occasionally, we used the system on mouse samples but these recordings were not included in the results.

[R1.10] In Fig. 2, why are only 2 of the 4 experts shown?

[A1.10] *The validation was performed on the annotations of 2 experts out of 4, indeed. For the training set generation the annotators were asked not to annotate the same images. However, for the validation set it is needed that the same images are annotated by multiple annotators (for the inter-expert accuracy) or by the same annotator multiple times (for the intra-expert accuracy). As the validation set was generated later in time and due to personal changes in the laboratories, we could not include annotations from all 4 people in this set.*

[R1.11] The performance analysis needs to be spelled out in more detail. To obtain true positives, false positives etc, the ground truth needs to be known. What *exactly* was the “ground truth” used for the precision and recall calculations? Is it the superset of the expert annotations? Did no experts cover the same dataset in the initial dataset? I am still unclear on this, despite reading the extended description in the Supp Material.

[A1.11] *Thank you for pointing out the lack of this important data. The validation dataset contained expert annotations that were not used for the training process and was annotated by one person. In the training dataset, no experts annotated the same images. However, for the intra- and inter-expert accuracies, two experts annotated the same selected image stack again after three months. Furthermore, for the performance analysis of the model, we have extended the validation dataset with more images and repeated the measurement due to concerns raised by the reviewer later in R1.13. We have modified the main text to contain this information and now it can be found in Results, Cell detection subsection. The table below shows the experts and their annotations on the validation set (0-not annotated, 1-annotated, V-part of the validation set, A-part of the intra-expert set, R-part of the inter-expert set).*

	Stack 1	Stack 2	Stack 3
Annotator 1	1VAR	1V	1V
	1A		
Annotator 2	1AR	0	0
	1A		

[R1.12] Fig. 2f-g: The purpose of this panel is to apparently show the drift in the cell as the pipette is lowered into the tissue. However, the figure needs a lot more clarification. What is the reasoning behind the template not being taken at the 0th image position and what is the numbering exactly if not an image position with respect to the template? Is it with respect to just the middle image? Then the standard deviation is taken for the entire difference image, which then can be used to show the drift in z-axis. The plots don't clearly indicate that it is

showing the standard deviation which should be shown. There are no error bars on this either. In the right-hand plot, the standard deviation for the position of the template image is comparable to the -2 and -1 positions on the left plot. Would be nice to show a control plot where there is not pipette movement and image stacks are taken repeatedly and compared to the same template image. This would show if the standard deviation of the difference images really show anything at all.

[A1.12] We thank the reviewer for the careful examination of the plot. There are several points mentioned which we address separately.

- *Figure clarification: We have made adjustments in the figure and its caption to clarify how the stacks are taken. Before the tracking is started, the template image is taken. Then, from time to time, small image stacks are taken of the cell such that the most recent focus position will be the 0th position, and the stack will contain a few more slices below and above it. For the first few stacks it is usually true, that the 0th slice is the most similar to the template image. In other words, the template image is taken at the 0th image position, but it is more correct to say that initially the image stack is created around the 0th position (where the template image was taken). Afterward, as the cell moves due to tissue deformation, the focus position can change and the previous statement will not stand anymore. It is important here that the template image is not updated throughout the process. Fig 3f (previously Fig 2f) shows a case where the stack is taken around the most recent focus position (indexed as 0th) and the algorithm detects that the cell has moved down. This results in lower metric values (shown in the plot) and the new focus position will be updated to 1 micrometer less than the previous value.*
- *Standard deviation plots: We have added axis titles to the plots (Fig 3f-g).*
- *Error bars: The two panels show individual examples, where the plots contain the exact metric values of the similarity of the images, thus error bars are not applicable.*
- *Control plot: We thank the reviewer for the idea of the control plot. We have performed the test and added the results to Supplementary Information: Cell tracking system. The conclusion is that the algorithm successfully detects in 89.17% of the cases when the cell is stationary, and that false detection of upward and downward movements are the same.*

[R1.13] “A central part of the method is the detection of single neurons in label-free 3D images using deep convolutional neural networks reaching super-human precision.”. This is a bold claim, considering that the precision of experts 1 and 2 in Figure 2d are higher than the precision of the detection algorithm. No evidence of super-human precision is demonstrated, and to do so would require eg a fluorescent label for ground truth, ie. It is not something that even could be demonstrated using the approach that has been taken – leaving aside that the claim seems to be false according to the authors’ own presentation.

[A1.13] We agree with the reviewer that our claim is not well-grounded and removed it from the text. Although the F1-score was higher for the algorithm than for the intra- and inter-expert tests, we now agree that this is not enough to make such a statement. We have analyzed what could lead to such confusing values and concluded that the validation dataset

might not be large enough. We have extended the validation image set (305 images in total) to have a better comparison. Furthermore, we have made a comparison of further deep learning algorithms. Updated results show that inter-expert F1-score is marginally higher than the best achieved by deep learning (FasterRCNN).

[R1.14] Regarding the trajectory, what angle is taken relative to the horizontal? This is an important piece of data.

[A1.14] In most of our measurements the mentioned angle was -33.14 degrees. We have updated the main text with this value. In the early development phase, sometimes we have adjusted the pipette and used our system with up to +/- 10 degrees difference. The calibration protocol automatically determines this value along with many others that are necessary for coordinate system transformation between the stage and pipette spaces. The complete list includes;

- 6 rotational angles, e.g. from the horizontal plane or the rotation from the X axis of the stage coordinate system in the horizontal plane.*
- 3 variables that indicate if the "forward" direction on a given pipette axis is a positive or negative movement in the stepper motor controller software.*
- 2 position vectors that store the stage and pipette positions at the time when the calibration was initiated.*

When the calibration protocol is executed, these values are written to the configuration file under config/vistool_config.xml. The actual values can be found in the XML under the <id>pip</id> identifier. However, the operator does not need to be aware of these values or remember them.

[R1.15] Why is recording quality measured only by R_s , rather than using R_{in} , which is more common? Are the 4 cells with R_{in} above 200 MOhms really acceptable?

[A1.15] R_s is an abbreviation for „series“ or „access resistance“ are synonymous and equal with the electrode resistance. Input resistance (R_{in}), however, describes the total resistance observed by the amplifier; it is therefore equal to the sum of the membrane resistance (R_m) and the electrode resistance (R_s), with the former generally dominating the sum. R_m is measured by the amplitude of the voltage change caused by injected steady current which is proportional to the number of open ion channels within the neuronal membrane. R_m above 200 MOhms is common especially among interneurons (Gouwens, Sorensen, Berg et al, 2019, Nat Neurosci.) and it can be relatively high (above 150 MOhm) in human superficial pyramidal cells (Kalmbach et al., 2018, Neuron). Note, that in the manuscript we use membrane resistance values instead of input resistance values therefore we amended abbreviation 'R_in' to 'R_m'.

[R1.16] Fig 4b should also have scale bars (not just depending upon looking down to 4e)

[A1.16] We have updated the figure to contain scale bars in both panels.

[R1.17] Fig 4 panels c and d are swapped relative to the caption

[A1.17] We are grateful to the reviewer for pointing out this mistake, we have adjusted the caption.

[R1.18] Penultimate paragraph, “integrated into” would work better than “portable to”.

[A1.18] We have changed the text accordingly.

[R1.19] Please use either indentation or paragraph spacing to separate paragraphs, it is very difficult to read otherwise.

[A1.19] We have added paragraph spacing to the main text and the supplementary materials as well for better readability.

[R1.20] The final paragraph begins with a rather ambitious aspiration. I do not believe that there is sufficient information in the DIC image to do this.

[A1.20] We agree with the reviewer that this sentence is ambitious and that it would not be possible to determine the phenotype. However, we see a chance that some groups could be detected, e.g. interneurons and pyramidal cells could be distinguished, which would have been our first step. We admit not being specific enough here and decided to remove this sentence.

Reviewer #2 (Remarks to the Author):

General Remarks

[R2.1] The study by Koos et al. (Automatic deep learning driven label-free image guided patch clamp system for human and rodent in vitro slice physiology) presents a hardware and software toolbox for automated patch-clamp recordings in in vitro acute slice preparations from rodent and human brain. The authors use deep learning for cell detection and a subsequent pipeline for pipette localization, tracking and ultimate accessing the neuron through whole-cell configuration. This manuscript, is part of a series of efforts in the neuroscience field for automating the laborious process of recording the electrophysiological properties of cells, intrinsic and synaptic, as well as obtaining the morphological

characteristics of the recorded neuron after filling it with a dye (typically biocytin or neurobiotin). In this case the authors also added mRNA profiling by aspirating the cell nucleus and cytoplasm after recording to detect the expression of key genes using digital PCR. Overall, the efforts presented in the manuscript are timely and aim at optimizing and streamlining a series of processes to present an end-end solution for performing high-throughput electrophysiological and molecular analysis of single cells in unstained tissue samples. Unfortunately, despite the great lengths that the researchers have gone to for this work, the study has the drawback that both the usage of deep learning for cell detection and the automation of patch-clamping have been implemented before independently, also in publications some of which the authors are citing (for ex. Kodandaramaiah et al Nat. Protocols 2016; Wu Q et al. J. Neurophysiology 2016; Suk SJ et al. Neuron 2017 for automated patching and Ounkomol C et al. Nature methods 2018; YoungJu Jo et al. arxiv 2018 for label-free cell and intracellular organelle detection). Nevertheless, the combination of methods that the authors have put together, from the usage of AI to detect unlabeled cells on brain slices and subsequently, not only electrophysiologically record these cells, but also reveal their morphology and some of their genetic makeup has not been shown before.

[A2.1] We are thankful to this reviewer for the detailed and constructive analysis and review of the work we present. We agree with the major point that both patch clamp automation and deep learning were proposed earlier, but as this reviewer pointed out the combination of these methodologies resulted in a unique technique.

Major issues:

Regarding the Machine Learning part:

[R2.2] Although a big focus of the study (as indicated in the title as well) is targeted in using deep learning for cell detection, the study unfortunately lacks a survey of existing cell detection techniques developed through deep learning-based methods. A quantitative comparison should be provided with other deep learning-based methods to show that the algorithm the authors have used performs better than other cell detection techniques (e.g. Iqbal et. al. Sci Reports. 2019, Zhou, Zhi, et al. Brain informatics 2018, Falk, Thorsten, et al. Nature Methods 2019), if trained on the same dataset.

[A2.2] We agree with the reviewer that our manuscript lacks such a survey and have addressed this issue. We have compared our method (Caffe-DetectNet) to other recent detection algorithms. Three further architectures were deeply analyzed, Faster Region-based Convolutional Neural Network (FRCNN-Resnet50); Darknet-ResNeXt; and Darknet-YOLOv3-SPP models and described them in the updated supplementary on cell detection. The FRCNN model (used also in Iqbal et. al. Sci Reports 2019) outperformed the other three, which had similar performance to each other. Since the patch clamping measurements were performed using the DetectNet model, we left its description in the main text, however, we have integrated the FRCNN model into the software. Now it can be changed in the configuration file. We remark that Falk, Thorsten et. al. Nature Methods 2019 describes the well-known U-Net architecture which can be used for segmentation tasks, well

applicable in bioimage problems. However, our approach is based on bounding boxes of the cells and not segmentation, thus we could not include a comparison with U-Net and other segmentation approaches.

[R2.3] Also, it would help the reader if a detailed explanation of the deep neural network architecture is included, and the same goes for an explanation of why DetectNet would perform well on unlabeled bright field images of neurons, as compared to other deep learning and non-deep learning-based cell detection tools (e.g. Iqbal et. al. Sci Reports. 2019, Zhou, Zhi, et al. Brain Informatics 2018, Falk, Thorsten, et al. Nature Methods 2019).

[A2.3] We have extended the description of the deep learning architecture which now can be found in the main text. Furthermore, as we mentioned in A2.1, we have performed a comparison to various deep learning models. We also put detailed discussions, figures and tables describing all four examined architectures in the “Deep Learning Model Comparison for Cell Detection” supplementary document.

[R2.4] Even though based on the accuracy score presented in Figure 2d, the algorithm seems to outperform human experts, the reviewer is of the opinion that neuron detection/localization (bounding box) is not optimal to get a 3D representation of the cell body for patching; if the neuron was segmented in 3D (also possible in 2D then digitally turned into 3D) a higher accuracy would probably be achieved (e.g. Januszewski, Michał, et al. Nature Methods 2018).

[A2.4] We agree with the reviewer that bounding boxes are not optimal representations of cells in 3D and believe that a proper segmentation could increase the success rate of the system. We have tried the seminal work of Januszewski et al, and other novel 3D deep architectures eg. Weigert et al. 2020 (StarDist), etc, however, we found that the segmentation quality of neuron cells on our DIC images was not satisfactory, and for automated patch clamping, the extent and center of the cell is sufficient. Therefore in this work we concentrate on detection networks and segmentation is out of the scope of this paper. 3D segmentation in DIC microscopy is our future plan.

[R2.5] It is not clear from the manuscript how much (what is the ratio) of the human annotated dataset was used for training and validation? Was there any cross-validation performed? A precision-recall curve is missing, as is a quantitative plot that shows the error rate of the network while training. It is not clear what kind of neuron images were used for training and testing, potentially the majority of the training and testing data is drawn from the same set hence the F1 score seems higher, which could simply be the result of overfitting.

[A2.5] The training was performed on 265 stacks, while the validation was performed on 3. The connection between how these stacks were used for evaluating the models and the experts (intra and inter) are shown in A1.11. However, the validation not only tested the deep learning algorithm but the 3D merging strategy of the bounding boxes as well. We have

updated the main text accordingly. We have paid attention to splitting the dataset so that images from the same stack go to the same set. We have analyzed how the F1 score could be higher for the algorithm and concluded that this is an issue with the validation set. Now we have extended the validation set and this value became lower. We do not state anymore that the algorithm outperforms human annotators.

We have included precision-recall and ROC curves in the updated supplementary for each model.

[R2.6] According to the authors, pre-trained weights of ImageNet were used, so I suppose transfer learning was applied. How many layers in the network were frozen during training? It would be good if the authors provided an explanation why GoogLeNet weights trained on ImageNet would work well for detecting neurons, given that ImageNet only contains natural images. Do images of unlabeled neurons in a slice have same features (statistics) as natural images (e.g. person)?

[A2.6] We indeed applied transfer learning. Earlier work from other labs (eg. Caicedo 2019 Nat Meth.) and our lab (Hollandi et al. 2020, Suleymanova et al. 2018, Moshkov et al. 2020) successfully apply pre-trained networks on natural images to single-cell detection and segmentation. The reasoning here is not trivial, we speculate that basic image feature representations (such as edges, simple shapes, color co-occurrences) are well preserved across these domains and can be a strong basis to solve more complex bioimaging problems. We did not freeze any layers to provide the opportunity to the network for slight changes in the early layers. We have also tried training without pre-trained weights (random initialization), but it did not give better results.

Regarding the electrophysiology/anatomy/mRNA part:

[R2.7] When the nucleus is harvested, the morphology of the cell usually cannot be maintained. What was the success rate for obtaining both?

[A2.7] During this project n=28 (n=8 rat and n=20 human) cytoplasm or nucleus were harvested from neurons patch-clamped by DIGAP with $R_s < 100$ MOhm. In 8 cases there was no anatomical recovery of the recorded cell, 5 cells recovered partially without an axon. In most of the cases (n=15) at least the axonal arborizations were observable and 10 of them (35.71%) had full anatomical (soma, dendrites and axon) recovery.

[R2.8] An access resistance of 30MOhm is high for whole-cell patch-clamp recordings. It may be that the intrinsic electrophysiological properties of the cells are not that affected by this high R_s , but the synaptic events would. Have the authors tried to record synaptic events using their system? Can the system change between current and voltage clamp automatically?

[A2.8] *We did not record evoked synaptic events from neurons with our present system. For that we will need two more extensions: i) to develop our system to perform automated patch clamp procedure with two or more pipette to do simultaneous recordings from cell pairs and ii) improve the number of patch-clamp recordings done with low series resistance. Indeed, to measure synaptic currents properly the ideal series resistance would be around 15 MOhm or at least <20 MOhm. To reach such quality we will have to refine the membrane seal phase which yet lacks the experimenter's intuitiveness to establish the ideal case.*

We used HEKA EPC amplifier system with HEKA PatchMaster software. The PatchMaster software offers batch communication opportunity to control the amplifier externally with other software. Our system uses this type of control for switching between current clamp and voltage clamp modes.

[R2.9] How stable were the recordings? The authors have a log system which registers all the values at any given point. How much did the access resistance change over time and how much time could the authors keep the cells stable and healthy?

[A2.9] *The cells were usually held in whole cell configuration at most for 15 minutes to protect neuron viability for further anatomical or RNA analysis. This time is enough for the biocytin to diffuse within the neuronal arborization. During this period the series resistance (Rs) was continuously monitored by the user but it was not recorded. The log system is recording values only during the hunting and sealing phases.*

Therefore - in order to answer this question - we conducted a separate set of experiment with n=9 neurons (n=1 pyramidal and n=8 interneurons) from rat somatosensory cortex in the same manner as described in our manuscript, to calculate series resistance from recorded membrane currents responded to short subthreshold voltage steps. We were able to keep five cells in whole cell mode for up to ~1 hour the other four only for a shorter period. The average time of experiments until the recording configuration could be maintained was 2729.9 ± 1104.2 s (min: 928 s, max: 3825 s). Averaged Rs values were at the start of the recording: 30.67 ± 14.98 MOhm (min: 21.91 MOhm, max: 69.79 MOhm). Rs values at the end of each recording were: 37.24 ± 18.13 MOhm (min: 20.46 MOhm, max: 74.47 MOhm).

[R2.10] In figure 5:

It would be good if the authors showed more examples of the reconstructed morphology of the recorded cells or at least images of z-projections in the supplementary figures. In this reviewer's experience a success rate of 80% in the recovery of morphology is very high indeed. This is even more surprising given that the authors also suck up the nucleus of the cell for RNA analysis.

[A2.10]

We prepared images of cells recorded with DIGAP system and inserted them in a separate supplementary document. We thank the reviewers for pointing out the 80% recovery data as it was incorrect and left mistakenly from an earlier version of the manuscript. We apologize for that. Indeed, the recovery rate was lower 40.9% for full and 27.5% for partial (missing either soma, axon or dendrite) for <30 MOhm samples. Among the harvested cells with

Rs<30 MOhm, we found 53.8% and 15.4% full and partial recovery. We corrected the results section accordingly.

[R2.11] In the Farago N et al. paper that the authors reference, the digital PCR method was used to quantify many genes that have specific biological functions, such as the delta subunit of GABA_A receptors, slc2a4 and microRNAs. Here the authors show the expression level of only two genes, which encode for proteins that are quite generic. It may be that these genes would have also been detected if aspiration of the extracellular debris was occurring instead of the cell nucleus. Have the authors performed a control experiment where they purposefully aspirate extracellular material to compare with the results they get in Figure 5b? The inclusion some more specific genes would be help assess how well this automated part works.

[A2.11] In the present work we performed mRNA expression analysis of single neurons according to the workflow presented in Faragó et al. (2013) and Faragó et al. (2016) to assess the efficacy of the automated patch system vs. manual patching in producing samples. To assure the best reference to published results, we chose two of the genes of intracellular and membrane proteins for mRNA expression analysis. The mRNA copy numbers presented in our manuscript are in correspondence with previous results although direct comparison can not be made due to dissimilarities in species and neuron subtypes therefore we have no clue to assume extracellular contamination. Please note that the number of copies rules out extracellular contamination which would mean low copy numbers of these genes.

The reason we showed dPCR results was to show that automation of the patch process might be successfully combined with cytoplasm harvesting. Since the step of aspiration of cytoplasm is out of the automation - as we clarified this in the answer to reviewer #1 - therefore we feel that including dPCR experiment with extracellular material would unreasonably drift our manuscript to a molecular biology study.

[R2.12] Finally, it would be nice if the authors presented a zoomed out video clip that would showcase how this system looks like, including the microscope, amplifier, automated manipulator etc, as well how it works once a slice is put down the chamber.

[A2.12] We have included a new annotated video (Supplementary Video 3) that shows the hardware setup as well as the screen capture of the patch clamping process.

Minor Issues:

[R2.13] There is no line numbering provided so as to include it herein, but the text where changes may be needed is provided below in italics.

[A2.13] We have added line numbering to the new version of the manuscript.

[R2.14] Please consider rephrasing the sentence: The quantitative and qualitative efficiency of single-cell patch clamp procedure is highly determinant for every follow up measurements including anatomical reconstruction and molecular analyses.

[A2.14] We have changed the mentioned sentence to: "Obtaining high-quality and numerous electrophysiological recordings from individual neurons is crucial for subsequent morphological and transcriptome analysis."

[R2.15] Please change: Recently, patch clamp technique has to Recently, the patch clamp technique has

[A2.15] We have accepted the reviewers suggestion.

[R2.16] Regarding the sentence: Blind patch clamping' moves the pipette forward in vivo
Blind patch clamping was first done in vitro and only later performed in vivo

[A2.16] We have utilized the suggested sentence and modified the previous one.

[R2.17] Please delete "of" in the sentence: electrophysiological measurements strongly correlates to that of made by a trained

[A2.17] We thank the reviewer for pointing out this grammatical error. We have modified the text accordingly.

[R2.18] It is stated that the arrows in Figure 2e are yellow, but they are white in color

[A2.18] Indeed, we thank the reviewer for noticing. Now we just write "arrows" as the color has no relevance.

[R2.19] Please change the word "were" to "was" and "vacuum" to "suction" in the following sentence: intracellular content of the patched cells were aspirated into the recording pipette with gentle vacuum applied by the pressure regulator unit (-40 mBar for 1 min, then -60 mBar for 2-3min, and finally -40 mBar for 1 min).

[A2.19] We have changed the text accordingly.

Reviewer #3 (Remarks to the Author):

[R3.1] The article is focused on using deep learning for recognition of DIC images of label-free neurons in brain slices for further automatic patch-clamp. Patch-clamp is the main electrophysiological technique for single-cell recordings. It is the primary methodology for the analysis of electrical properties of neurons and other electrically excitable cells. Several recent publications from different laboratories have described automated patch clamp systems, both blind and image-guided in slices and in vivo. These different systems are able to use computer vision libraries to detect pipette tips, fluorescently labeled cells, adjust their patch path, wash and reuse the pipettes, patch multiple cells at the same time.

The key innovation of the authors' paper is the use of deep learning first to train their neural network and then recognize the images of neurons in brain slices acquired using DIC optics. This cell recognition is then combined with an automatic patch-clamp. The software is written in Matlab with external deep learning library calls. Overall, the use of deep learning to recognize cells in DIC optics is important for the field, even though all other parts of the process have been published previously. However, there are several problems that need to be addressed, including a poor description of the deep learning part of the paper. I am enthusiastic about this paper, provided the issues below are addressed, and the manuscript is revised.

[A3.1]

We are grateful for the enthusiasm of this reviewer as well as her/his detailed and constructive comments. We agree that parts of this process were published earlier, but we give here a unique combination of these methodologies as a working pipeline. Below, we give a detailed answer to each question/point raised by this reviewer.

There are several issues that need to be addressed:

[R3.2] First, it is not clear if the patching process is fully automatic or if the interference of a human may be required, and if it is, how often. The corresponding statistics are not sufficiently extensive; only the success examples vs. failures are mentioned. What is the percentage of fully automatic vs. attempts with human interruption vs failures to patch?

[A3.2]

Patching process can go fully automated with the possibility for the experimenter to stop the ongoing process at any time and choose to restart the phase or jump to the next phase.

In an earlier phase of the development to detect failures of automatic trials we registered the rate of full automatic run and human interaction in n=62 consecutive experiments. 56.4% (n=35) of them were conducted fully automated without the experimenters interaction including n=16 successful and n=19 unsuccessful electrophysiological records. Fully automated runs with unsuccessful electrophysiological record caused by typically a failure in membrane suction phase causing improper connection with the cell. Note that the suction/sealing step is quite uncontrollable, for example the success of membrane seal strongly depends on the nucleus distance from pipette tip during membrane suction.

In n=27 (43.5%) experiments human intervention was done because of the following reasons:

- *The system did not change from sealing phase to breakin phase, although the resistance change as trigger was given (n=2). This error was caused by a rare incident caused by the delayed send of signal due to occupation of the PatchMaster*

program maintaining other processes e.g. calculating holding current. In these cases the experimenter switched phase manually.

- *During the break in phase the membrane did not open enough or closed back (n=4). In these cases the sucking of membrane was interrupted and proceeded by the experimenter.*
- *The system did not change from hunting phase to sealing phase although the resistance change as trigger was given (n=2). This was caused by the deformation of tissue. For example if the advancing pipette is too close to a blood vessel and touches it a large portion of the tissue can be pushed forward by the vessel and pipette. This is detected as a progress suspension by the system therefore it remains in the same phase. In these cases the experimenter switched phase manually.*
- *The pipette did not hit the cell (n=17). This problem occurred typically at samples with uneven surface, or a large blood vessel sticks to the pipette wall which then as advanced forward deformed the tissue in an irregular manner. In these cases the experimenter moved the pipette manually to the cell, then started the sealing phase.*
- *The system can not detect if the membrane suddenly breaks in without forming giga-seal during the sealing phase (n=2). In these cases the experimenter stopped the sealing phase and started the e-phys recording manually.*

Furthermore, we have included two new videos, Supplementary video 1 and 2 that show screen captures of two successful automatic patch clamping processes.

[R3.3] The hardware configuration in Figure 1 doesn't actually show any meaningful hardware, for example, which patch-clamp amplifier, how it's connected to the National Instruments board. What kind of signals the NI board receives, and what commands does it send and where?

[A3.3] We have separated the two parts of previous Fig. 1 and replaced the hardware part with photos of the actual devices we used, this can be found in Fig 2. Furthermore, we have updated the Results, Hardware development and control section with information on how we measure the resistance and control the amplifier. As the NI board is connected mostly to the pressure controller parts we refer to the Supplementary Information: Pressure Regulator Setup for a detailed wiring diagram. Finally, we refer also to A3.4 as these questions are highly related.

[R3.4] How do the authors control the amplifier and send/receive information? Is the output impedance then received by the NI board?

[A3.4] We used HEKA EPC amplifier and PatchMaster software (HEKA) for measuring the electrophysiological signals. PatchMaster offers a "batch file control" protocol for controlling the amplifier from other applications. Our system uses this strategy for switching between current clamp and voltage clamp modes, to set the holding potential, to start the recording etc. We used this as a one-direction communication in which the amplifier is the receiver.

The electrophysiological signals were digitized and recorded by PatchMaster and were directly monitored with the NI board (the current monitor output of the amplifier was connected to an analog input channel of the NI board). From this monitored signals our

system calculated the total resistance of the pipette. If it was reasonable (i.e. the pipette resistance increased because the pipette tip hit the target cell) the DIGAP software sent batch commands to the amplifier to perform fast capacitance compensation, start the recordings and so on. The updated Figure 2. contains the information flow in the system.

[R3.5] The detailed information and which commands are used by the authors to switch between voltage/current clamp modes, measure impedance, inject voltage command needs to be documented in a supplementary file in addition to the bitbucket or github or another online open-source repository.

[A3.5] Our basic stimulus set and batch communication script is available in a separate document uploaded to <http://bitbucket.org/biomag/autopatcher/> and as a new supplementary information. Briefly, we used short negative current pulses for series resistance monitoring during pipette maneuvering and membrane sucking phase and 800 ms long step cycled with increment. In some cases we used stimulation waveform set defined by Allen Brain Institute Data Portal 'Electrophysiology Overview' technical white paper (<http://help.brain-map.org/display/celltypes/Documentation>)

[R3.6] How the z-focus is changed is not clear. This information is also important, especially in consideration of wider adoption in the neuroscience community. Different slice rigs may have different methods to change the z-focus. How do the authors control the z-focus from the software, and how does it integrate with both pipette detection, cell detection, z-stack acquisitions? A more detailed description should be both in the main and supplementary figures.

[A3.6] The microscope we used was modified by Femtonics Ltd. (Hungary) and includes a motorized Z axis, provided with an API to query or set its position. We agree that different setups have different methods to change the z-focus. Although we cannot provide a working solution for every microscope, we have implemented our software such that the controller classes are inherited from abstract base classes. Thus, if a new device is to be used (a different motorized Z axis in this case), then only a new child class is to be written that implements a few abstract methods and our software remains functional.

Setting the z-focus is utilized in many parts of the system. Generally, an image stack is acquired for every automated step, including the cell detection, pipette detection, or cell tracking in the Z axis. Of course, there are a few cases when manual operation is required, for example when looking for the sample top position or finding the tip of a fresh pipette. This is done with the controller wheel also provided by the manufacturer.

We have updated the Methods, Hardware setup, and Results, Hardware development and control sections to clarify this issue.

[R3.7] The graphical user interface(GUI) of the software is shown only in the supplemental file. There should be at least a separate main figure showing the main elements of GUI with

the detailed annotation. This figure should also include the GUI for initiating the cell recognition process.

[A3.7] We have added a new figure (Fig. 5) in Results, Software section to include screenshots of the GUI. It includes four panels, including the built-in labeling tool, the monitoring window, the main window when browsing the detected cells, and the automated laboratory notebook module.

[R3.8] In the video shown, one can see the patching process, but not the deep learning-based recognition of the cells with further successful targeting. It would be useful to include a video of the cell recognition process.

[A3.8] We have added another video (Supplementary Video 2) that is a screen capture of the whole patch clamping process. It includes the part when an image stack is acquired, the cells are detected, and the operator browses them.

[R3.9] In the video of the patching process, one can see the changes in resistance, but not the pressure. It is consequently not clear when the negative pressure was applied to break-in. How much suction was applied, etc. Is this how the process usually works? Or is there a separate pressure window which was not shown in the video? Probably a better video could be used, showing the average patching process.

[A3.9] Our previous video (Supplementary Video 1) is for a quick demonstration of the system's capabilities. The new Supplementary Video 2 shows a full, successful visual patch clamping process. Unlike the resistance value, the pressure value is not shown historically for the last few seconds. However, the actual value is always visible in the top right Pressure panel in the Blind patcher window. The pressure values shown in the representative examples (Fig. 4 in the main text and the new Supplementary Information: Representative examples) were extracted from the log files.

[R3.10] There is no diagram of the cell recognition->automatic patching algorithm. Consequently, it's not clear what happens after what. At least, I was not able to easily find the algorithm diagram.

[A3.10] We have updated Fig. 1 with a schematic that shows the time dependence of the steps of visual patch clamping.

[R3.11] The Figure 1a panel represents a series of confusing icons that can hardly be seen and are not an informative algorithm diagram.

[A3.11] As already mentioned in A3.10, we have updated Fig. 1 by adding arrows between the icons to demonstrate the order in which the different steps are performed.

[R3.12] The paragraph describing cell recognition and specifically deep learning, including training, is very brief and needs to be vastly expanded. There should be more information, more details about how the cells are recognized, what happens to the recognized cells in each z-stack cross-section, how they are then combined, etc. This is probably the biggest innovation of the paper, yet there is almost no information about how this is done.

[A3.12] We have integrated the related Supplementary Information into the main text, Results, Cell detection section. Furthermore, we have added a description on how the detections are combined along the z-axis.

[R3.13] Figure 2 describing the algorithms, and the cell detection module is incomplete. Specifically, Figure 2b has the model mentioned, but which model? Is it a convolutional network or maybe LSTM? This is not clear, yet this information is critical and needs to be obvious from arguably the most important figure in the paper.

[A3.13] We have updated Fig. 3 (previously Fig. 2) which now contains “Convolutional Neural Network” instead of just “Model”. Furthermore, we have included an extensive description of the algorithm in the Results, Cell detection section.

[R3.14] The information about calling Caffe libraries from Matlab is only available in the supplemental information, with only a brief mention of the requirement of a separate computer with GPUs for training vs. recognition. This kind of information about the core innovation of the paper needs to be in the main text.

[A3.14] We have moved (and updated) the previous supplementary information on cell detection to the main text. Most luckily, due to recent Matlab updates it became possible to import Caffe networks and calling Caffe libraries is not required anymore, thus we have removed this part from the text. Furthermore, we have performed a comparison of deep learning algorithms (new supplementary document) and implemented a Faster R-CNN algorithm besides Caffe. The detection systems can be changed in the main configuration file as described in Supplementary Information: Software usage and parameters, which also serves as a user manual.

[R3.15] A lot of other information about the setup is very hard to find. For example, the authors use the 40x water immersion lens. This needs to be in the first figure describing the hardware setup. What happens if one uses a 63x lens, will it work with the software?

[A3.15]

We have addressed this issue in multiple ways. We have reworked the figure of the hardware setup (Figure 2). Furthermore, we have updated the Results, Hardware

development and control, and Methods, Hardware setup sections. The former contains the description of hardware elements that are developed by us or just controlled by the software, while the latter contains the element descriptions that are used as-is.

Related to higher magnification objectives, we had access to a 60x objective and tested the cell detection algorithm. Below are some selected image-pairs of the same tissue regions, 40x on the left with highlighted regions of the 60x pair on the right. As can be seen, the cells are often undetected, or when detected, usually the confidence is lower. This is expected, as CNNs are often not scale invariant. The patch clamping procedure could be used after setting the correct pixel size in the software. This value is either given by the manufacturer or can be determined by a calibration sample. However, we have not performed such measurements with this objective.

[R3.16] How does the system operate when you switch between a lower magnification lens (10x?) used for the initial targeting of the brain slice. Do the authors even use one? It is not clear from the article. Is the initial calibration of the pipette and the derivation of the slice->pipette transformation function necessary at the low magnification? Is it done manually or fully automatically?

[A3.16] We have not tested our system with lower magnification objectives either and we think it would raise further problems. First of all, deep learning architectures are known not to perform well on small objects. Neurons with a 10x objective would look very small in the digital images compared to 40x and we do not expect even a re-trained algorithm to perform well. Moreover, even if there would be a reliable object detector it would be very hard to precisely target with the pipette so that it does not “slip off” of the neuron. As for the pipette calibration, it would be necessary when the objective changes. The most important would be to determine the pixel size in micrometers. The value which is given by the objective manufacturer usually does not consider other optics elements, including the focal length of the digital camera. Thus it should be determined manually, preferably with a calibration slide or microbeads. In our case for the 40x objective, Femtonics provided a good approximate for this value, but the final value (0.115 micrometer) was determined empirically.

[R3.17] There are several questions related to pipette detection. First, the authors cite only their own previous conference paper [21] while ignoring previous earlier work from other groups demonstrating successful automatic pipette tip detection.

[A3.17] We thank the reviewer for pointing out that this should be detailed. We are aware of previous approaches from other groups, some of which we already cited in the manuscript (Suk, HJ et al. 2017; and Wu, Q et al. 2016) but did not cover the pipette detection part. We have added a discussion about these and one other approach (Yang, R et al. 2014) in the Introduction. Unfortunately, we could not include them in the comparison due to differences in imaging conditions. In case of fluorescent modality, the inside of the pipette becomes visible which opens up new analysis possibilities, unlike in label-free imaging when only the edges of the pipettes are visible. As for the approaches that use label-free modality, they utilize a low magnification objective (4x) which results in a long depth of field (for details we refer to this Nikon page), and thus the edges of the pipette are sharp all over the image. As we had no access to a motorized/automated objective changer and our 40x objective has much lower depth of field, we decided to develop our own solution for the pipette localization problem for images where the edges of the pipette are sharp only in a rather small region.

[R3.18] Another confusion is related to the pipette tip detection. It is mentioned that the mouse clicking on the tip of the pipette is required for calibration. If this is correct, this may not represent the fully automatic detection of the pipette tip.

[A3.18] In the early phase of development we implemented only the manual tip update functionality. As the fully automatic detection is still somewhat slower, we decided to leave

the manual in as well. However, the fully automatic detection works as intended and presented, and the operator has the option to use either.

We would like to note that, in our terminology, pipette calibration and tip position update are two different actions. Calibration is the process of setting the pipette axes and calculating the transform matrix to the coordinate systems of the stage. This requires at least 4 tip detections, and even more when performed stepwise in an automated manner for higher precision. In this case it is much easier and faster to manually follow the pipette tip with the stage and the objective and set the new tip positions. Furthermore, it is only required before the first use of the system. On the other hand, tip position update is performed after a fresh pipette is inserted, or the current one is cleaned. In this case, the pipette is assumed to be close to its estimated position, has to be detected only once, and time difference compared to manual detection is not so significant.

[R3.19] Figure 3a is also confusing. The path of the pipette seems very strange for an algorithm. I assume the idea here is to demonstrate a patch attempt with an obstacle hit. There is no annotation of the stages 1,2,3 in Fig. 3a. I assume 3 represents a complicated avoidance path, but this is a very complicated path. Is this performed by a human to re-position the pipette? It looks from the schematic as if some of the path includes lateral movements within the slice? Not sure.

[A3.19] Indeed, Fig. 3a demonstrates the obstacle avoidance algorithm. The trajectory was reconstructed from a successful measurement where an obstacle was hit. Indeed, step 3 indicates lateral movement in a spiral path which is based on the referenced paper (Stoy, WA et al., 2017). We have further annotated the image and added a description of the steps.

[R3.20] Fig. 3b. There are several non-regular changes in the pipette pressure during the approach, which are strange? Were there any commands performed by the algorithm to apply positive/neutral pressure? Also, after the start of the sealing process, again, several step-like changes in the negative pressure. Again, without the algorithm or a description of their state-system, it is not clear what is going on here.

[A3.20] We thank the reviewer for pointing out that more details are required regarding the patch clamp phases. We have added more information to the Results, Automated patch clamping steps section. During the whole process the pressure should be kept at a few (4-6) predefined values, thus the changes visible in the figure are irregularities when the controller was unable to readjust it fast. This is mostly due to pulling back the pipette in the beginning of the obstacle avoidance part, when the pipette leaves an empty space in the tissue and the pressure escapes easily.

[R3.21] There need to be more representative examples of successful and unsuccessful patches (both path and diary files with resistance and pressure measurements).

[A3.21] We have created a new supplementary document that contains 9 successful and 6 failed representative examples. Each contains 2 plots, the reconstructed pipette trajectory in 3D, and the triplet of pipette depth, pressure, and resistance values. These attempts originate from the measurements included in the main text. The image stacks are not saved automatically when the patching process is started, thus identical figures to Fig. 4 in the main document could not be created. We tried to select and include diverse attempts in the new supplementary: e.g. pipette got adjusted by cell tracker, an obstacle was hit, the attempt was (re)started from the break-in phase, the break-in was successful in a few/lot of attempts, etc.

Minor issues:

[R3.22] Some of the citations are not precise. For example, “ ... Object detection of neurons in label-free tissue images is challenging [24]...” 24 represents a paper by the authors, which performs image recognition on the astrocytes stained using immunohistochemistry (IHC). It would probably be more appropriate to say something like: ... we have previously demonstrated the identification of labeled cells using deep learning following immunohistochemistry.

[A3.22] We have rewritten this part of the text and the cited sentence is not present anymore.

[R3.23] There needs to be more extensive discussion how different hardware can be used with the authors' DIGAP system. Also, there needs to be more discussion of how different deep learning frameworks can be potentially used with the software.

[A3.23]

We have added a discussion in the Results, Hardware development and control section on how to use different hardware with the software than what is described. Furthermore, we have integrated another deep learning framework (FRCNN) in the software and describe in Supplementary Information: Software usage and parameters how it can be selected. This is also an example of how arbitrary frameworks can be integrated.

Reviewers' Comments:

Reviewer #1:

Remarks to the Author:

The authors have comprehensively addressed my concerns.

Reviewer #2:

Remarks to the Author:

General Remarks:

"Automatic deep learning driven label-free image guided patch clamp system for human and rodent in vitro slice physiology"

I would like to thank the authors for putting some effort in addressing the points that were raised. I continue to maintain that their approach to automate the in vitro recording of unlabeled cells and hence the electrophysiological, anatomical and potentially also molecular characterization of human neurons or other non-genetically tractable species is useful. I am nevertheless not entirely convinced about some of the technical aspects of this study especially the way deep learning is implemented for the goals of the project, also because it is the most novel of the aspects of their pipeline.

Regarding the Deep learning approach

Even though the authors have now included more information about their method and also tested other Deep Learning architectures on cell detection, they still do not present a very strong argument for pushing the deep learning approach as part of their pipeline for the following two main reasons.

Firstly, the method that they ultimately decide to go with, DetectNet, does not have a good performance (F1 score ~60%). In the first round of reviews the authors were asked to test alternative deep learning methods to assess the performance of their network. After running 3 other networks on their datasets (2 being closely related, Darknet XX), they demonstrate that Faster RCNN gives significantly better results than DetectNet. The authors nevertheless decide not to focus their paper on Faster-RCNN, because they had performed all their electrophysiological recordings with DetectNet and also because of the time complexity of Faster-RCNN. The reason though that time may be an issue, as tested by the authors, is because they have used ResNet as the backbone architecture. As has been shown in Iqbal, et. al. Sci. Reports. 2019, a good level of precision can be achieved in much less time if a simple backbone architecture is built with only a few convolutional layers. If the argument against this network is the time sink, the authors could aim at a high level of precision with less time complexity for their application. For this it'd be important to just adapt a simple backbone architecture of Faster R-CNN and then compare its performance with ResNet-based Faster-CNN and their existing DetectNet architecture. A performance comparison plot of accuracy (mean average precision curve) and time complexity will be useful.

Secondly, as also mentioned in the first review, automating cell detection for whole-cell patching is not really an object detection problem, it is rather an instance-based object segmentation issue. Therefore, even expanding the detected bounding box to several frames in z direction will not solve the issue of 3D segmentation since the structure, shape and size of neurons are extremely diverse so the precision of bounding box in z-depth#1 will be off as we move deeper into the tissue. The authors mention in the text that they have tried 3D segmentation methods, but did not

get good results from them “We have tried the seminal work of Januszewski et al, and other novel 3D deep architectures eg. Weigert et al. 2020 (StarDist), etc, however, we found that the segmentation quality of neuron cells on our DIC images was not satisfactory”.

Nevertheless they do not show any of these results or the performance these achieve and claim that for their purpose only the cell detection network they have used is sufficient and works well. Unfortunately, the data that they present though suggests that their proposed method does not work optimally and has a poor performance based on the F1 score. An accuracy between 55-60% is already close to a random chance which can probably easily be achieved by a naive linear classifier, henceforth, there seems to be little advantage of using a deep learning-based network for this purpose.

Besides these two main points, there are a couple of other important ones listed below:

1) Having now provided more information on the implementation of their network, the reviewer finds it surprising that the training was done on 265 stacks and the testing on 3 (the authors report that this corresponds to almost 300 images). If this is correct, it means that the results only report on about 1% of the total dataset. The standard protocol is to use a train/test split of 80/20 or 70/30 % (randomly shuffled and cross-validated), which is used to measure the generalized performance of a machine learning-based method. With 3 stacks, the reviewer thinks it is unlikely that the results will be generalizable, due to over-fitting.

2) In addition, the number of epochs is stated as 6 (Line#225-227). Please check if this is a mistake, since the network cannot learn any features with a epoch size of 6. Sometimes thousands of epochs need to be used in order to train a network for a reasonable performance.

3) To the question whether the authors use transfer learning, they reply positively. Nevertheless, when asked about the number of layers that are frozen in the network, they replied that “We did not freeze any layers to provide the opportunity to the network for slight changes in the early layers”. These two replies are somewhat contradictory. If all the network layers are frozen during training then no transfer learning is applied.

For all the above issues and even though deep learning is presented as probably the most important aspect of the novelty of the pipeline presented in this paper, I would propose that it is somewhat toned down throughout the text and/or the low performance and the need to improve on the approach should be clearly stated. The focus of the paper could instead be on an effort towards automating the cell patching system, also using deep learning for cell detection, instead of focusing too much on deep learning, which seems to be the weaker part of the paper.

Regarding the electrophysiology.

In this part I will follow the specific answers of the authors

Regarding the previous point [R2.8]

In terms of the detection of synaptic events I did not imply that the authors should implement automated whole-cell paired recordings by introducing a second pipette. I was referring to spontaneous synaptic events by recording in continuous mode, either in voltage or current clamp. Since the idea of the approach that the authors propose is the automatic recording of electrophysiological properties of neurons and many labs are performing standard recordings of spontaneous synaptic events in control versus genetically modified tissue for example, it may be nice if the authors showed some synaptic events (if they have already collected them) and also comment on that.

Regarding the previous point [R2.9]

Thank you for adding these new recordings. Again it would be good if the authors commented on

their new recordings in the text somewhere, letting the readers know that the cells could be kept for up to 1 hour according to their new efforts.

Reviewer #3:

Remarks to the Author:

The manuscript is much improved.

I have no other issues.

Review Answers

Our response to the comments from the reviewers

Below, we provide a detailed summary of the comments made by the reviewers. Each comment is followed by our response (*denoted in italic font*). We have responded to every comment [Rx.y] from the reviewers and made a genuine effort to address all concerns [Ax.y]. In order to clearly denote where changes have been made to the manuscript, we have **highlighted changes from the previous version of the manuscript in red**.

REVIEWER COMMENTS

Reviewer #1 (Remarks to the Author):

[R1.1]

The authors have comprehensively addressed my concerns.

[A1.1]

We are grateful to this reviewer for his time, effort, and the critical comments that we believe resulted in higher quality work.

Reviewer #2 (Remarks to the Author):

[R2.1]

General Remarks:

"Automatic deep learning driven label-free image guided patch clamp system for human and rodent in vitro slice physiology"

I would like to thank the authors for putting some effort in addressing the points that were raised. I continue to maintain that their approach to automate the in vitro recording of unlabeled cells and hence the electrophysiological, anatomical and potentially also molecular characterization of human neurons or other non-genetically tractable species is useful. I am nevertheless not entirely convinced about some of the technical aspects of this study especially the way deep learning is implemented for the goals of the project, also because it is the most novel of the aspects of their pipeline.

[A2.1]

We thank the reviewer for carefully examining our study again. We have performed further deep learning tests to address the issues that are raised. The reviewer has pointed out in

R2.4 that the dataset should be split better so that the validation set becomes bigger. To be able to run every new test necessary (including parameter search, comparisons, and cross validation), we have decided to revise the dataset and select a representative subset (35 stacks) of human samples that were imaged in the later stages. We have used this smaller dataset for this round of review, and in the manuscript we indicate when this subset is used. Please find our detailed answers below.

Regarding the Deep learning approach

[R2.2]

Even though the authors have now included more information about their method and also tested other Deep Learning architectures on cell detection, they still do not present a very strong argument for pushing the deep learning approach as part of their pipeline for the following two main reasons.

Firstly, the method that they ultimately decide to go with, DetectNet, does not have a good performance (F1 score ~60%). In the first round of reviews the authors were asked to test alternative deep learning methods to assess the performance of their network. After running 3 other networks on their datasets (2 being closely related, Darknet XX), they demonstrate that Faster RCNN gives significantly better results than DetectNet. The authors nevertheless decide not to focus their paper on Faster-RCNN, because they had performed all their electrophysiological recordings with DetectNet and also because of the time complexity of Faster-RCNN. The reason though that time may be an issue, as tested by the authors, is because they have used ResNet as the backbone architecture. As has been shown in Iqbal, et. al. Sci. Reports. 2019, a good level of precision can be achieved in much less time if a simple backbone architecture is built with only a few convolutional layers. If the argument against this network is the time sink, the authors could aim at a high level of precision with less time complexity for their application. For this it'd be important to just adapt a simple backbone architecture of Faster R-CNN and then compare its performance with ResNet-based Faster-CNN and their existing DetectNet architecture. A performance comparison plot of accuracy (mean average precision curve) and time complexity will be useful.

[A2.2]

We agree that simpler backbones can often be used to achieve good quality results and that we might have put our efforts too much on detection quality and less on usability due to hardware limitations. Now, we have tested the FRCNN architecture with simpler backbones as well. First, we have tried the DeNeRD model that was used in Iqbal et. al. 2019. The training was performed for 100 epochs with decay of 0.2 every 10 epochs and initial LR=1e-3. The results were F1=42.59, P=50.42, R=37.86, and the requested curves are shown in Fig. 1 below. Unfortunately, these were rather low even compared to the DetectNet model, thus we tried to use another light-weight network, MobileNetV2. This backbone is much less complex than ResNet-50 and also known to provide good results (Bianco et. al. 2018). We have trained this network with the same settings. The results were F1=60.93,

$P=55.13$, $R=71.47$ (Fig. 2). This model is competitive and we have included it in the software.

Figure 1: precision-recall and ROC curve of the FRCNN-DeNeRD model.

Figure 2: precision-recall and ROC curve of the FRCNN-MobileNetV2 model.

[R2.3]

Secondly, as also mentioned in the first review, automating cell detection for whole-cell patching is not really an object detection problem, it is rather an instance-based object segmentation issue. Therefore, even expanding the detected bounding box to several frames in z direction will not solve the issue of 3D segmentation since the structure, shape and size of neurons are extremely diverse so the precision of bounding box in z-depth#1 will be off as we move deeper into the tissue. The authors mention in the text that they have tried 3D segmentation methods, but did not get good results from them “We have tried the seminal work of Januszewski et al, and other novel 3D deep architectures eg. Weigert et al. 2020 (StarDist), etc, however, we found that the segmentation quality of neuron cells on our DIC images was not satisfactory”.

Nevertheless they do not show any of these results or the performance these achieve and claim that for their purpose only the cell detection network they have used is sufficient and works well. Unfortunately, the data that they present though suggests that their proposed method does not work optimally and has a poor performance based on the F1 score. An accuracy between 55-60% is already close to a random chance which can probably easily be achieved by a naive linear classifier, henceforth, there seems to be little advantage of using a deep learning-based network for this purpose.

[A2.3]

With the assumption that the somata of the neurons are convex objects, aiming for the center of their bounding box should be rather close to the centroid of their accurately determined shape/contour. We agree that aiming for their centroid is theoretically more correct and believe that it can improve the ultimate success rate of automatic patch clamp systems. However, given the complexity difference between object detection and instance segmentation tasks, and that there are no other publicly available annotated DIC tissue dataset, we believe our choice is obvious and useful for the community. As generating annotations for 3D segmentation would be very time consuming, we think that other approaches could be used to still aim for the cell centroid based on object detection results. Lee J. et. al. (2018) developed a segmentation and tracking method in DIC tissues that expects a manually given seed point and can be used for automatic patch clamp systems. We think the integration of this algorithm can be an improvement of our system in the future.

We would like to make up for not showing the mentioned results previously. As our annotations are for object detection task (i.e. bounding boxes), we have used networks trained on the Kaggle DSB 2018 fluorescence images for nuclei segmentation. The DIC images were reconstructed by our previous DIC reconstruction algorithm (Koos 2016), that were the input of StarDist-2D (Fig. 3) and StarDist-3D (Fig. 4). The result instances are thresholded at 0.8 confidence. There are still many false detections in the image, and the contours of the correctly found neurons are rather rough (as it can be expected for this type of training).

Figure 3: Result of StarDist-2D on a reconstructed DIC image. Left: original DIC image. Right: Result of StarDist-2D on the phase-reconstructed image.

Figure 4: Result of StarDist-3D on a reconstructed DIC image. Left: original DIC image. Right: Result of StarDist-3D on the phase-reconstructed image.

The reviewer mentioned that 55-60% accuracy is close to a random choice and thus the use of deep learning is not justified. We do not know what accuracy could be reached with hand-crafted feature analysis and linear classifiers (although we have started with a similar approach many years ago when we started this project, but it did not prove to be useful), we would like to note that a random choice in the object detection task would give much worse accuracy than 50%, possibly something close to 0%. This is because objects in the image can appear in any number, anywhere, and with various sizes. Furthermore, we compare 3D bounding boxes which makes this task even more complex. The random choice is 50% accurate only in binary classification problems.

Besides these two main points, there are a couple of other important ones listed below:

[R2.4]

1) Having now provided more information on the implementation of their network, the reviewer finds it surprising that the training was done on 265 stacks and the testing on 3 (the authors report that this corresponds to almost 300 images). If this is correct, it means that the results only report on about 1% of the total dataset. The standard protocol is to use a train/test split of 80/20 or 70/30 % (randomly shuffled and cross-validated), which is used to measure the generalized performance of a machine learning-based method. With 3 stacks, the reviewer thinks it is unlikely that the results will be generalizable, due to over-fitting.

[A2.4]

We agree with the reviewer that using 1% of the dataset for testing is rather low. Our idea was that this way we can compare every algorithm with multiple manual annotations. Since the network was used for hundreds of recordings we do not think it overfit, but still agree that the metrics can be misleading. Note, that the annotators had to annotate the same stacks multiple times for the intra- and inter-accuracies and increasing the size of the test set would be very time consuming. Because of this and assuming that the best architecture (FRCNN) will be used from now on, we have performed a new training and cross validation with FRCNN on the subset of the full dataset as mentioned in A2.1. The cross-validation showed that the network, when trained for 20 epochs, gives an F1-score of 65.33% which is close to

the previously reported 65.83%. Please find the results in the cell detection supplementary material.

[R2.5]

2) In addition, the number of epochs is stated as 6 (Line#225-227). Please check if this is a mistake, since the network cannot learn any features with a epoch size of 6. Sometimes thousands of epochs need to be used in order to train a network for a reasonable performance.

[A2.5]

We have checked that the training was run for 6 epochs and the number is correct. At the moment, we are running a training using 2 NVIDIA GeForce RTX 2080 Ti cards for 20 epochs on this dataset, but 1 epoch takes 21.5 hours and it will take 18 days to finish. We will evaluate the quality of this model and make it available with the source code on Bitbucket if it performs better. We plan to do the same for 100 epochs (approx 3 months runtime). Please note, that we finetuned the network pretrained on ImageNet and that the 265 stacks resulted in 12600 images in the training set. These together allowed us to use a low number of epochs, but without initialization we agree that we might have needed hundreds or even thousands of epochs. However, we have performed a further test where we checked how the training performs after 6, 20, and 100 epochs on the smaller dataset (mentioned in A2.1). The quality is similar to what was given for the previous test set of 3 stacks. 20 epochs proved to be marginally the best choice, but the network is competitive after only 6 epochs. Please find the detailed results in the cell detection supplementary material.

[R2.6]

3) To the question whether the authors use transfer learning, they reply positively. Nevertheless, when asked about the number of layers that are frozen in the network, they replied that “We did not freeze any layers to provide the opportunity to the network for slight changes in the early layers”. These two replies are somewhat contradictory. If all the network layers are frozen during training then no transfer learning is applied.

[A2.6]

We are somewhat confused by the last sentence, as we wrote that we did not freeze any layers but left all of them trainable. Initializing a network using the parameters acquired at the end of another training on a slightly different task is a very common approach in deep learning. Depending on how many data points we have in the target dataset we can choose to freeze some of the layers or no layers at all. If no layers are frozen during the training, the process is called fine tuning that is the sub-area of transfer learning. We believe that freezing the layers is optional and since we transferred the knowledge from another task in the materialization of the learned weights, our approach can be called transfer learning. We refer to the seminal work of Yosinski et. al. (2014) for the terminology that is aligned with our manuscript.

[R2.7]

For all the above issues and even though deep learning is presented as probably the most important aspect of the novelty of the pipeline presented in this paper, I would propose that it is somewhat toned down throughout the text and/or the low performance and the need to improve on the approach should be clearly stated. The focus of the paper could instead be on an effort towards automating the cell patching system, also using deep learning for cell detection, instead of focusing too much on deep learning, which seems to be the weaker part of the paper.

[A2.7]

We thank the reviewer for pointing out that deep learning is over expressed in the manuscript. We have made changes in the text and included ideas in the Discussion on how the cell detection could be improved.

Regarding the electrophysiology.

In this part I will follow the specific answers of the authors

[R2.8]

Regarding the previous point [R2.8]

In terms of the detection of synaptic events I did not imply that the authors should implement automated whole-cell paired recordings by introducing a second pipette. I was referring to spontaneous synaptic events by recording in continuous mode, either in voltage or current clamp. Since the idea of the approach that the authors propose is the automatic recording of electrophysiological properties of neurons and many labs are performing standard recordings of spontaneous synaptic events in control versus genetically modified tissue for example, it may be nice if the authors showed some synaptic events (if they have already collected them) and also comment on that.

[A2.8] Although we could detect spontaneous events we did not perform long term continuous recordings for analysis of spontaneous events. Therefore we have performed longer continuous voltage clamp recordings from a fast-spiking basket cell and a pyramidal neuron to record spontaneous excitatory postsynaptic currents (sEPSC). We detected spontaneous events from 30 seconds long recordings using the Neuromatic tool (Rothman and Silver, 2018) for Igor (Wavemetrics). We collected $n=866$ and $n=24$ sEPSC from the basket and pyramidal cell, respectively. In the figure below, the left sides of the panels show a representative 450 ms and 1.7 sec long part of the recordings (panel A basket cell; panel B pyramidal cell). Red dots denote detected sEPSC events. Right sides show the averaged sEPSCs with SD.

We indicated these informations in the manuscript by inserting the following sentence:

During our measurements we were able to detect spontaneous postsynaptic events in the entire length of the recordings.

A

B

[R2.9]
Regarding the previous point [R2.9]

Thank you for adding these new recordings. Again it would be good if the authors commented on their new recordings in the text somewhere, letting the readers know that the cells could be kept for up to 1 hour according to their new efforts.

[A2.9]

We have accepted the reviewers suggestion and now included the followings in the Results section of the manuscript:

Once the whole cell configuration was formed cells were usually held at most for 15 minutes to protect neuron viability for further procedures. To test the stability of whole cell configurations we executed a separate set of experiments and found that half of the trials (n=5 out of 9) could be kept up to 1 hour. The average time of experiments during the recording configuration could be maintained was 2729.9 ± 1104.2 s (n=9, min: 928 s, max: 3825 s).

Reviewer #3 (Remarks to the Author):

[R3.1]

The manuscript is much improved.
I have no other issues.

[A3.1]

We thank the reviewer for reading our changes and for his time, effort, and critical comments that we believe have improved the quality of our manuscript.

Reviewers' Comments:

Reviewer #2:

Remarks to the Author:

The reviewer would like to thank the authors for the extra effort they put into the manuscript and for addressing the points raised concerning the deep learning and experimental parts. I have no further issues.